# *Schistosoma mansoni* treatment reduces HIV entry into cervical CD4+ T cells and induces IFN-I pathways

Sergey Yegorov [1,10], Vineet Joag [1,11], Ronald M. Galiwango[1], Sara V. Good [2,3], Juliet Mpendo[4], Egbert Tannich [5], Andrea K. Boggild[6,7], Noah Kiwanuka[4,8], Bernard S. Bagaya[4,9] & Rupert Kaul [1,6]

*Schistosoma mansoni* (*Sm*) infection has been linked with an increased risk of HIV acquisition in women. Therefore, defining the mechanism(s) by which *Sm* alters HIV susceptibility might lead to new HIV prevention strategies. Here, we analyze the impact of standard *Sm* therapy in HIV-uninfected *Sm*+ Ugandan adult women on genital HIV susceptibility and mucosal and systemic immunology. Schistosomiasis treatment induces a profound reduction of HIV entry into cervical and blood CD4+ T cells that is sustained for up to two months, despite transient systemic and mucosal immune activation and elevated genital IL-1α levels. Genital IFN-α2a levels are also elevated post-treatment, and IFN-α2a blocks HIV entry into primary CD4+ T cells ex vivo. Transcriptomic analysis of blood mononuclear cells post-*Sm* treatment shows IFN-I pathway up-regulation and partial reversal of *Sm*-dysregulated interferon signaling. These findings indicate that *Sm* therapy may reduce HIV susceptibility for women with *Sm* infection, potentially through de-repression of IFN-I pathways.

[1] Departments of Immunology and Medicine, University of Toronto, 1 King's College Circle, Toronto, ON M5S 1A8, Canada. [2] Genetics & Genome Biology, The Hospital for Sick Children, Peter Gilgan Centre for Research and Learning, 686 Bay St., Toronto, ON M5G 0A4, Canada. [3] Community Health Sciences, University of Manitoba, 750 Bannatyne Ave, Winnipeg, MB R3E 0W2, Canada. [4] Uganda Virus Research Institute –International AIDS Vaccine Initiative HIV Vaccine Program, 51/59 Nakiwogo Rd, P.O.Box 49 Entebbe, Uganda. [5] Bernhard Nocht Institute for Tropical Medicine, National Reference Centre for Tropical Pathogens, Bernhard-Nocht-Str. 74, 20359 Hamburg, Germany. [6] Department of Medicine, University Health Network, 200 Elizabeth Street, Toronto, ON M5G 2C4, Canada. [7] Public Health Ontario Laboratories, 661 University Ave, Toronto, ON M5G 1M1, Canada. [8] Department of Epidemiology and Biostatistics, School of Public Health, College of Health Sciences, Makerere University, P.O. Box 7072 Kampala, Uganda. [9] Department of Immunology and Molecular Biology, School of Biomedical Sciences, College of Health Sciences, Makerere University, P.O. Box 7072 Kampala, Uganda. [10] Present address: Department of Science Education, Faculty of Education and Humanities, Suleyman Demirel University, 1/1 Abylai Khan Street, Kaskelen, Almaty 040900, Kazakhstan. [11] Present address: Department of Microbiology, University of Minnesota, Minneapolis, MN 55455, USA. Correspondence and requests for materials should be addressed to S.Y. (email: sergey.yegorov@mail.utoronto.ca)

The helminth *S. mansoni* (*Sm*) causes intestinal schistoso-miasis, a neglected tropical disease that affects an estimated 54 million people in sub-Saharan Africa[1], despite the recent expansion of mass treatment programs[2]. *Sm* inhabits the gastrointestinal vasculature of the human host and has a complex life cycle that consists of infection after contact with fresh water containing infectious cercariae, worm migration to the mesenteric microvasculature, and subsequent passage of parasite eggs into the gut lumen[3]. Although adult schistosomes themselves suppress immunity by modulating immune system components, including T helper (Th)1 and interferon signaling[4], their eggs induce inflammation that damages the gut mucosa and surrounding tissues[3]. In addition, egg deposition induces a strong Th2 response that can cross-regulate Th1 responses[5,6]. These diver-gent effects of *Sm* infection are thought to impair host immune defenses against other pathogens, and may enhance susceptibility to human immunodeficiency virus (HIV)[7,8]. The strongest epi-demiological signal for the latter is seen in women[9–11], although not in all cohorts[12,13]. Furthermore, large cohort studies in *Sm*-endemic Uganda demonstrated lower HIV infection risk in people with a history of schistosomiasis treatment[12,13]. While one potential mechanism for enhanced HIV susceptibility is the impairment of systemic antiviral defenses by helminthic immu-nomodulation, it is also plausible that *Sm* egg-induced inflam-mation of the gut mucosa activates common mucosal homing pathways[14] with enhanced CD4+ T cell trafficking to genital sites of HIV exposure through expression of the mucosa-homing integrin α4β7. Indeed, higher levels of this integrin on blood CD4+ T cells have recently been associated with increased HIV acquisition[15], and may directly enhance cellular HIV susceptibility[16].

Therefore, to test the hypothesis that *Sm* treatment of infected women would reduce genital HIV susceptibility we performed a prospective clinical study in a region of Uganda with a very high prevalence of schistosomiasis. We demonstrate that *Sm* clearance substantially reduces cervical CD4+ T cell susceptibility, and boosts both mucosal and systemic IFN-I antiviral responses. These findings help to elucidate the impact of *Sm* infection and its treatment on antiviral immunity and HIV acquisition, and may point the way to novel strategies to reduce HIV transmission in the region.

## Results

**Study participants and parasitological assessment**. In this registered clinical trial (ClinicalTrials.gov #NCT02878564), we recruited adult schistosome-infected HIV-negative women from communities around Lake Victoria living in the radius of ~2.8 km from the lake (demographics in Table 1 and Supplementary Table 1) into a prospective study of genital HIV susceptibility. The diagnosis of schistosomiasis was made based on the urine circulating cathodic antigen (CCA) test, and 36 consenting, *Sm*-infected women were provided with a standard dose of oral praziquantel for *Sm* treatment at Visit 1 (V1); of these, two participants were subsequently excluded due to infection by *Chlamydia trachomatis* and/or *Neisseria gonorrhoeae* (Fig. 1a). At baseline, the study participants had no signs of organomegaly by palpation and were within normal range for basic biometric, co-infection and socio-behavioral characteristics (Table 1). Addi-tional diagnostic tests were performed for schistosomiasis spe-ciation and to assess schistosomiasis burden (Supplementary Fig. 1), since the CCA test is highly sensitive[17] but does not permit schistosome speciation. *Sm*-specific serology was done in all participants at all three study visits, *Sm*- and *Sh*-specific PCR was performed in all participants at baseline and two months post-praziquantel therapy, and Kato-Katz microscopy was

performed at baseline. These species-specific but less sensitive tests found no cases of *Sh* infection, and confirmed *Sm* infection in 24/34 women (71%); this "*Sm* PCR/serology+ subset" con-sisted of participants who were either positive for both *Sm* ser-ology and PCR (20/24), *Sm* PCR alone (1/24) or serology alone (3/24) (Supplementary Table 3). In addition, 12/34 women were *Sm* PCR/serology+ and had detectable *Sm* eggs on stool micro-scopy (35%, "Kato-Katz+ subset", Fig. 1b); no *Sh* infections were detected by urine microscopy. Eosinophilia, a non-specific mar-ker of helminth infection, was present in 21.6% of the partici-pants; two individuals had hookworm and/or *Trichuris* (Table 1), and no cases of *Ascaris*, *Strongyloides* or *Trichostrongylus* infec-tion were detected. Rates of injectable hormonal contraceptive (HC) use in the cohort were low (5.9%, Table 1).

**Post-treatment diagnostic outcomes**. Follow-up visits at one and two months post-praziquantel treatment (V2 and V3, respec-tively) were attended by 29/34 (85%) and 24/34 (71%) of parti-cipants (Fig. 1a). Post-treatment visits occurred at the same point (± 2 days) in the participants' menstrual cycle as the baseline visit; participants who were not cycling were followed up after 28 ± 2 days. The median duration between the baseline (prazi-quantel administration) visit and follow-up visits V2 and V3 was 28 (IQR 27–30) and 56 days (IQR 54–59), respectively. The efficacy of schistosomiasis therapy was assessed via CCA score and *Sm*-specific serology at post-treatment visits. Successful praziquantel therapy is expected to result in a reduction (though not complete clearance) of circulating schistosome antigens[17] and IgG antibodies against *Sm* eggs[18]. Consistent with successful

### Table 1 Participant characteristics at study enrollment

| Characteristic (*n* = 34 participants) | N (%)/median (IQR)/ mean (range) |
|---|---|
| Median age (IQR) | 24.50 (21.75–30) |
| Mean body mass index (range)^ | 24.11 (16.66–34.41) |
| Blood hemoglobin, g/dl (IQR)^ | 13.4 (12.08–14.13) |
| Eosinophilia*, n (%) | 8 (21.6) |
| CCA scores, n (%) | |
| +1 | 25 (73.5) |
| +2 | 6 (17.6) |
| +3 | 3 (8.8) |
| *Sm* eggs detected in stool by Kato-Katz: | |
| Negative, n (%) | 22 (65) |
| Positive, n (%); median epg (IQR) | 12 (35); 192 (72–312) |
| Geohelminths detected in stool by Kato-Katz: | |
| Hookworm and/or *Trichuris*-positive, n (%) | 2 (5.8) |
| HSV-2 seropositive, n (%) | 16 (47.1) |
| Hormonal contraceptive use, n (%) | 2 (5.9) |
| DMPA, n (%) | 1 (2.9) |
| NetEn, n (%) | 1 (2.9) |
| Self-reported genital condition in past month**, n (%) | 9 (26.5) |
| Presence of BV (Nugent score 8–10), n (%) | 12 (35.3) |
| Menstrual cycle stage at baseline | |
| Proliferative (follicular), n (%) | 13 (38.2) |
| Secretory (luteal), n (%) | 16 (47.1) |
| Not cycling, n (%) | 5 (14.7) |
| Sex in last 3 days (PSA+), n (%) | 13 (38.2) |
| Reported condom use in last sex, n (%) | 4 (11.8) |

*IQR* interquartile range, *epg* eggs per gram (of stool), *DMPA* depot-medroxyprogesterone acetate, *NET-EN* norethisterone enanthate, *PSA* prostate-specific antigen
^normal body mass index range = 18.5–24.99 and blood hemoglobin median in women = 13.5 g/dl (http://apps.who.int/bmi/index.jsp); * eosinophilia in this study was defined as > 450 eosinophils per ul of blood; ** self-reported vaginal itching/discharge, pain on urination or abdominal pressure

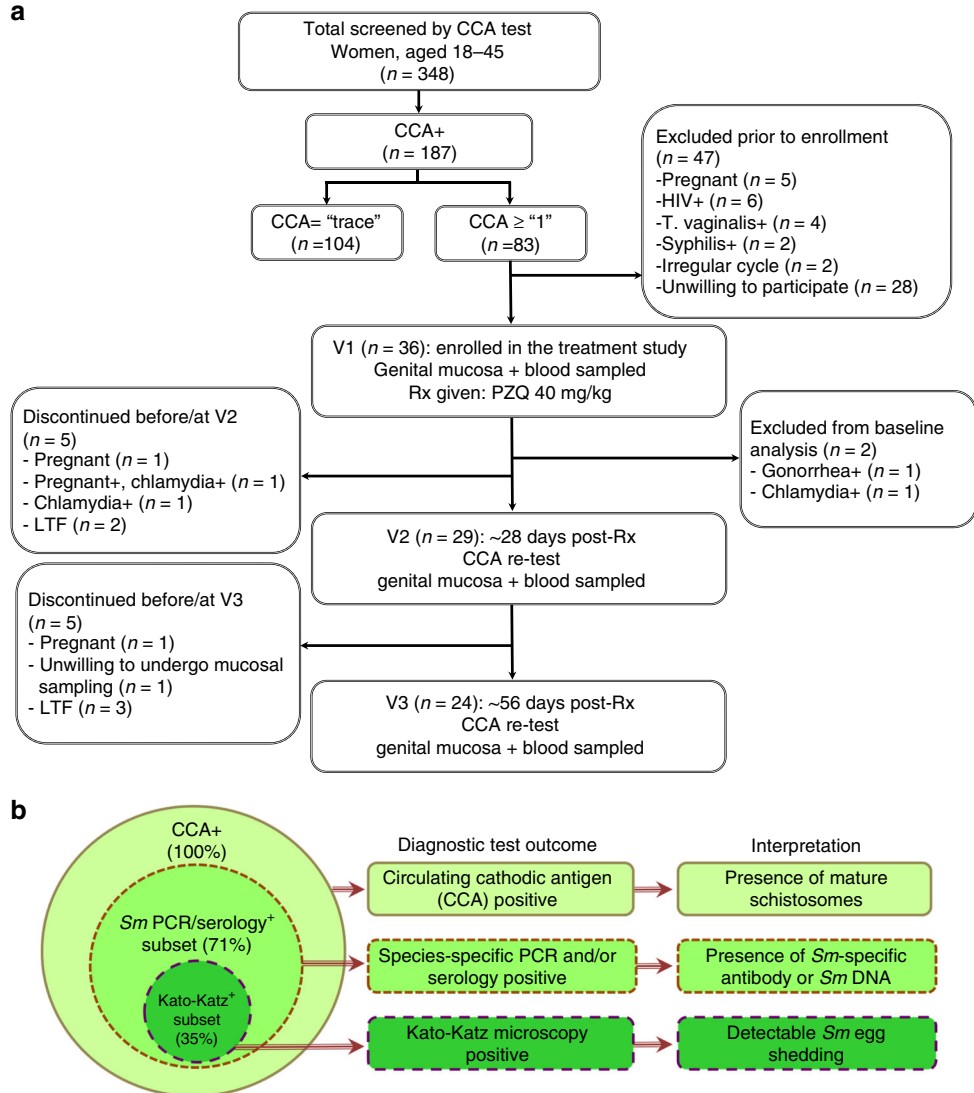

**Fig. 1** Overview of the study recruitment scheme and schistosomiasis testing outcomes. **a** Screening and recruitment flow chart. See Methods for detailed study inclusion criteria and Supplementary Table 2 for the CCA scoring scheme. **b** Proportion of all participants who were CCA-positive (34/34), were tested positive by *Sm* PCR and/or *Sm*SEA serology (24/34) or shed *Sm* eggs as determined by microscopic analysis of stool samples (12/34) at baseline analysis. See Supplementary Table 3 for details. Source data are provided as a Source Data file. CCA: circulating cathodic antigen; PZQ: praziquantel; *Sm*: *Schistosoma mansoni*; SEA: soluble egg antigen; LTF: lost to follow-up

treatment of *Sm*, a reduction of CCA scores was seen in 89.7% (26/29) and 91.7% (22/24) of participants at V1 and V2, respectively (median CCA change −0.5, $p < 0.001$, Supplementary Tables 4 and 5); there was also a significant reduction of *Sm*-specific antibody titers (Supplementary Fig. 2), as well as a reduced eosinophil count by two months post-treatment (Supplementary Fig. 2). In the majority (88%) of participants, *Sm*-specific antibody titers and PCR Ct values did not cross the pre-defined thresholds for positivity after treatment (serology OD > 0.2 and PCR Ct < 45). Marital status, herpes simplex virus type 2 (HSV-2) prevalence, bacterial vaginosis, contraceptive use and sexual behavior (as defined by prostate-specific antigen positivity) did not vary appreciably across study visits.

**Sm treatment reduced HIV entry into cervical CD4+ T cells.** The impact of schistosomiasis treatment on the entry of a clade A, CCR5-tropic HIV pseudovirus[16] into cytobrush-derived cervical CD4+ T cells, the pre-defined primary trial endpoint, was

assessed in all participants (Fig. 2a and Supplementary Figs. 3–4). HIV entry into cervical CD4+ T cells was reduced by 2.4-fold at one month post-treatment ($p = 0.001$) and remained lower at two months (1.6-fold reduction, $p < 0.001$; Fig. 2b, Supplementary Table 6). HIV entry into blood-derived CD4+ T cells, a secondary endpoint, was also reduced at both of the post-treatment visits (median fold reduction = 1.30 and 1.23 respectively, $p = 0.001$ and $p = 0.005$) (Fig. 2b, Supplementary Table 8). In addition to the reduced proportion of infected CD4+ T cells, there was also a trend to a lower number of HIV-infected CD4+ T cells per cytobrush (median fold reduction = 1.21; $p = 0.093$; Fig. 2b, Supplementary Table 8). Similar reductions in virus entry were seen in the *Sm* PCR/serology+ participant subset; in this group, there was also a significant reduction in HIV-infected cervical cell numbers (median fold reduction = 1.79; $p = 0.033$) (Supplementary Table 8). The reduction in viral entry was still seen in a sub-analysis, in which we excluded BV+ participants and those participants whose BV status changed between visits.

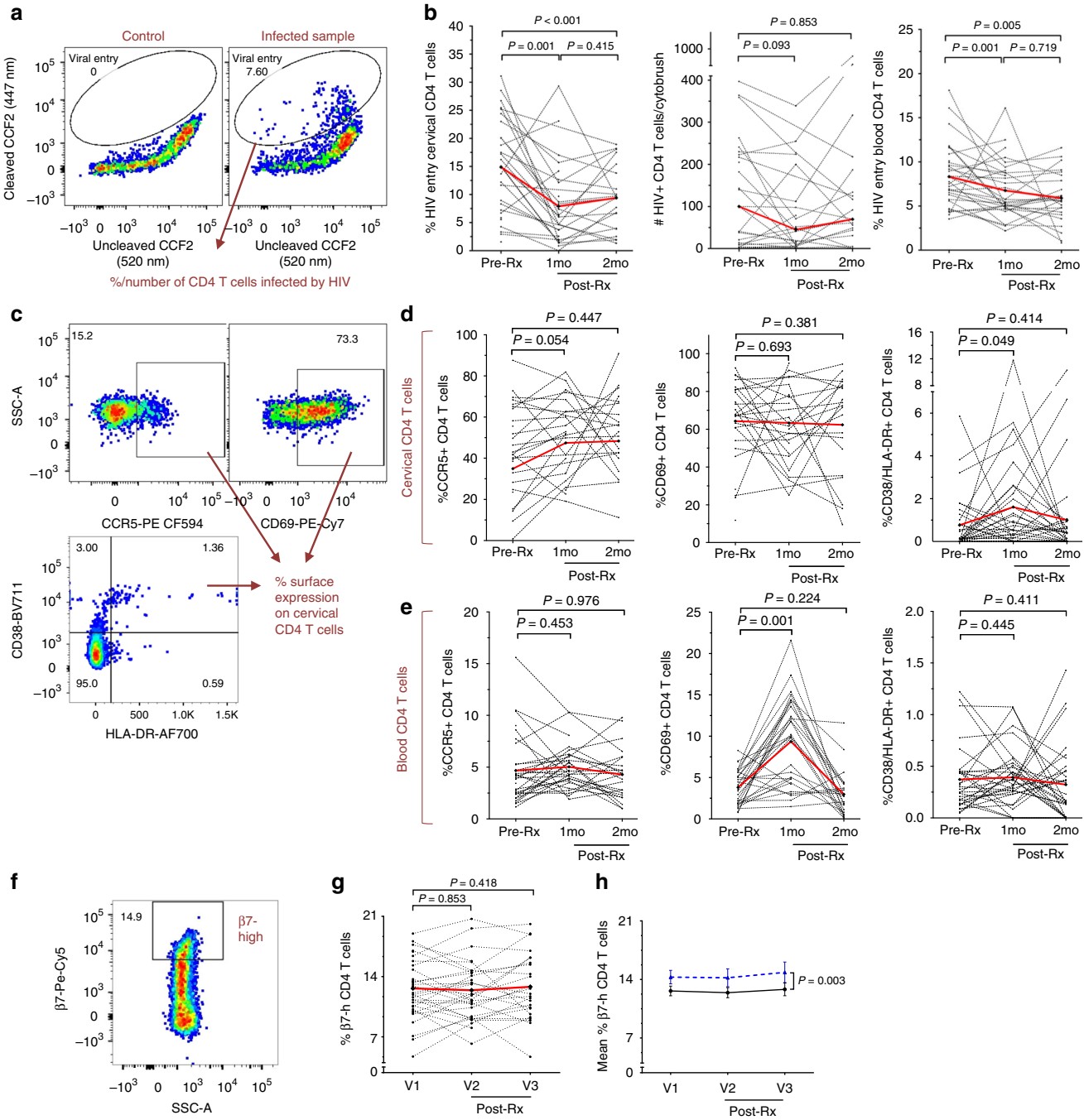

**Fig. 2** Cellular HIV entry, immune activation and β7$^{high}$ CD4+ T cell levels before and after *Sm* therapy. **a** Representative dot plots depicting HIV pseudovirus entry in cervical CD4+ T cells. Cells were pre-gated on single live CD4+ T cells (Supplementary Fig. 3). **b** Flow cytometric analysis of HIV entry into cervical (percentage and absolute numbers) and blood (percentage) CD4+ T cells. Fold changes and sample sizes for virus entry data are listed in Supplementary Table 8. **c** Representative dot plots depicting gating for CCR5+, CD69+ and CD38+/HLA-DR+ cervical CD4+ T cells. Cells were pre-gated on single live CD4+ T cells (Supplementary Fig. 3). **d, e** Changes in the percentage of cervical (**d**) and blood (**e**) CD4+ T cells positive for CCR5, CD69 and CD38/HLA-DR. Red lines denote mean percentage of cells positive for a marker at each study visit. Representative plots for **e** are shown in Supplementary Fig. 4. **f–h** Representative dot plot depicting the gating for integrin b7$^{high}$ blood CD4+ T cells. Cells were pre-gated on single live CD4+ T cells (Supplementary Fig. 4). **g** Levels of circulating b7$^{high}$ CD4+ T cells before and after schistosomiasis treatment depicted as individual participant data, where red line denotes mean β7$^{high}$ percentage of cells at each study visit. **h** Mean percentage of β7$^{high}$ CD4+ T cells in the CCA + (black line) vs. Kato-Katz+ (blue dashed line) participants. Significance was assessed by paired t-test (**b, d, e, g**) or (in **h**) repeated measures ANOVA (test of between-subjects effects with "Kato-Katz positivity" as factor, F = 10.86). In panels **b, d, e** and **g**: each dot represents an individual participant, dashed lines connect samples paired across the study visits, n = 23 and 20 for paired cervical comparisons, while n = 31 and 25 for paired blood comparisons at one and two months, respectively. Source data are provided as a Source Data file. *Sm: Schistosoma mansoni*

**Schistosomiasis therapy led to transient immune activation**. At the cellular level, HIV entry correlates with HIV co-receptor expression and cellular activation[19]. Therefore, we hypothesized that *Sm* treatment might have reduced cellular HIV entry through down-regulation of the HIV co-receptor CCR5 and/or reduced immune activation. Predefined secondary endpoints therefore included the expression on cervical and blood CD4+ T cells of the HIV co-receptor CCR5, the immune activation markers HLA-DR/CD38, and CD69. Unexpectedly, CCR5 expression on cervix-derived CD4+ T cells tended to increase after *Sm* treatment ($p = 0.054$), and CD38/HLA-DR expression increased significantly ($p = 0.049$) (Fig. 2d). These changes were not apparent in blood-derived CD4+ T cells, although systemic CD69 expression was transiently increased 2.4-fold one month after treatment ($p = 0.001$; Fig. 2e). The Kato-Katz+ subset demonstrated a higher level of post-treatment CD4+ T cell activation (Supplementary Fig. 5), and treatment-mediated increases in CD69 expression correlated with both baseline egg shedding ($R = 0.675$, $p = 0.016$) and CCA intensity ($p = 0.034$) (Supplementary Fig. 6). Total cervical T cell numbers tended to increase one month after *Sm* therapy ($p = 0.082$) and increased significantly ($p = 0.046$) after two months (Supplementary Fig. 7 and Supplementary Table 9), as did blood lymphocyte counts (Supplementary Fig. 2).

To further explore the effects of *Sm* treatment on inflammation and immune activation, we quantified a pre-defined panel of mucosal and blood cytokines (Supplementary Table 10 and Fig. 3). Genital inflammatory cytokine levels tended to increase (Fig. 3a–c), consistent with the treatment-induced cervical CD4+ T cell activation, and genital interleukin (IL)−1α levels were significantly elevated one month post-therapy in *Sm* PCR/serology+ (fold change = 1.71, $p = 0.003$) and Kato-Katz+ (fold change = 1.60, $p = 0.028$) individuals (Fig. 3b, c). In contrast, blood cytokine levels tended to fall after *Sm* treatment, with significant reductions in specific cytokines previously linked to active schistosomiasis[20,21], including tumor necrosis factor (TNF), IL-2, IL-10 and IFN-γ; these changes were not seen in schistosomiasis-negative controls treated with praziquantel (Supplementary Fig. 8). Overall, the impact of *Sm* therapy on cytokine levels was quite distinct in the blood and genital tract, likely reflecting compartmentalized differences in the timing and nature of treatment-induced immunological responses. However, contrary to our hypothesis, the clear reduction in cervical CD4+ T cell HIV entry that had been induced by praziquantel treatment was accompanied by a transient increase in mucosal immune activation, both at the cellular and tissue level, that directly correlated with the pre-treatment helminth burden.

**α4β7 + CD4+ T cell frequencies were unaltered by *Sm* treatment**. Next, we assessed whether the observed reduction in cervical HIV entry after *Sm* treatment resulted from a reduced frequency of circulating mucosa-homing α4β7 + cells, by examining the frequencies of circulating β7$^{high}$ CD4+ T cells, which are > 99% α4β7 + [22] (Fig. 2f). Treatment did not alter overall β7$^{high}$ cell frequencies in blood (Fig. 2g); although being *Sm* egg-positive at baseline was associated with an elevated β7$^{high}$ CD4+ T cell frequency (mean of 14.28% vs. 12.62%), this elevated frequency remained unchanged post-treatment ($p = 0.003$; Fig. 2h).

**HIV entry reduction was probably not due to a direct drug effect**. Praziquantel has minimal off-target effects and is rapidly cleared after oral dosing (plasma half life ≈ 2–4 h)[23], making it unlikely that reduced HIV entry at one and two months after treatment would be a direct drug effect. However, to assess this possibility we collected blood from four *Sm*-uninfected controls (three women and one man), at baseline and one month after the administration of empiric praziquantel due to a recent travel history. These individuals subsequently tested *Sm*- negative by all diagnostic tests (Supplementary Table 11), and blood-derived CD4+ T cells were assessed using pseudovirus assays and flow cytometry. Praziquantel administration, in the absence of schistosomiasis, did not tend to reduce HIV entry into blood-derived CD4+ T cells or alter systemic CD69+ CD4+ T cell frequency (Supplementary Table 12).

**Schistosomiasis therapy was associated with elevated IFN-α**. Mucosal immune activation would generally be expected to increase HIV susceptibility[24], but reduced simian-HIV acquisition despite mucosal immune activation was previously observed in macaque models of vaginal and systemic IFN-I administration[25,26]. IFN-I signaling is also stimulated by the *Sm* egg antigens that are released after praziquantel treatment[27–29], and IFN-I exhibits anthelminthic properties by suppressing *Sm* egg production and egg-induced granuloma formation in mice[30]. In addition, CD69 is induced on lymphocytes after IFN-I stimulation[31], and IL-1α is a major IFN-I signaling modulator[32]. Therefore we hypothesized that *Sm* therapy triggered a global proinflammatory IFN-I response, which induced immune activation but also exerted antiviral effects at the stage of cellular entry by HIV[33–36]. In keeping with this, genital interferon (IFN)-α2a levels were increased (fold change = 1.3, $p = 0.01$) one month after *Sm* treatment in both *Sm* PCR/serology+ and Kato-Katz+ participants (Fig. 3g). Furthermore, a trend to elevated genital and blood panIFN-α was seen one month after *Sm* treatment ($p = 0.068$); IFN-β was undetectable in most mucosal and blood samples (Supplementary Table 10).

**Exogenous IFN-α reduced *ex vivo* HIV entry into CD4+ T cells**. The effect of IFN-α2a on cellular HIV entry was then tested using blood lymphocytes from five *Sm*-uninfected donors: here, exogenous IFN-α2a directly reduced ex vivo HIV entry into CD4+ T cells (4.14-fold, $p = 0.043$) consistent with previous reports of IFN-I pathway-mediated HIV inhibition at the cellular entry stage[35,36]. Notably, exogenous IFN-α2a also increased CD69 and CCR5 expression (3.3 and 1.13-fold, respectively, $p = 0.043$; Fig. 3h, i and Supplementary Fig. 9) on blood CD4+ T cells. Thus, overall, *Sm* treatment increased genital IFN-α2a levels, while in vitro this cytokine induced cellular changes that closely mimicked our in vivo trial findings.

***Sm* treatment resulted in a systemic IFN-I induction**. To further explore the hypothesis that *Sm* therapy resulted in a global IFN-I induction, we performed a transcriptomic analysis on stored peripheral blood mononuclear cells (PBMC) collected from three participants before and after *Sm* treatment, who were randomly selected from the subset of Kato-Katz+ participants who had cleared their *Sm* infection at both one- and two-months post-therapy. The timing of sample collection for these participants was 27–33 days and 54–55 days at V2 and V3, respectively. One of these individuals tested positive for HSV-2 and BV; no other co-infections were detected in these participants (see Supplementary Table 14). To reduce the potential impact of inter-individual variation, we employed a novel computational approach, and performed a paired intra-individual analysis, which enhanced our statistical power despite a relatively small sample size of nine samples. RNA-seq analysis revealed that, compared to baseline, the number of differentially expressed genes (DEGs) one month after treatment was approximately double that seen after two months, with 18.7% DEG overlap (Supplementary Fig. 14), in keeping with the more substantial early post-treatment cellular and cytokine changes. IFN-I

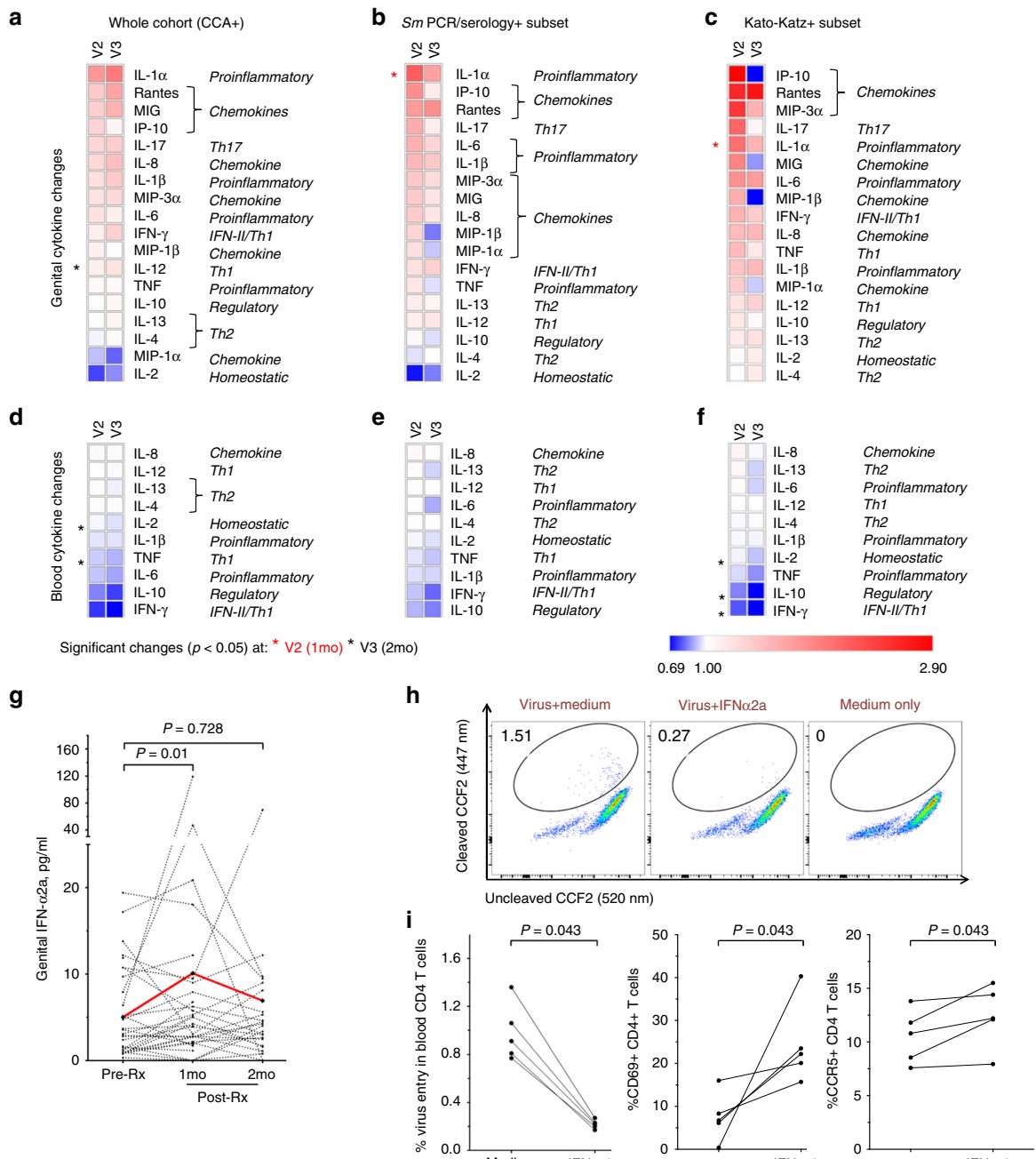

**Fig. 3** Genital and systemic cytokines before and after *Sm* treatment and IFN-α effects on cellular HIV entry. **a–f** Genital (**a–c**) and blood (**d–f**) cytokine level changes represented as the geometric mean of fold change of each cytokine at post-treatment visit (V2 or V3) over pre-treatment level in: (**a**, **d**) CCA + , (**b**, **e**) *S. mansoni* PCR/serology+ and (**c**, **f**) Kato-Katz+ participants. Cytokines were plotted in descending V2/V1 fold change order. Each cytokine was assigned a category (right of heatmap) based on their function or sub-type. Scale denotes fold changes ranging from 0.69 to 2.90 and where 1.0 is "no change". The numbers of participants with paired samples analyzed are in **a**: V2 = 31, V3 = 24; in **b**: V2 = 22, V3 = 17; in **c**: V2 = 12, V3 = 10; in **d**: V2 = 30, V3 = 25; in **e**: V2 = 22, V3 = 16; in **f**: V2 = 12, V3 = 10. See the Source Data file for raw cytokine concentrations for all participants at each study visit. **g** Plots depicting individual participant changes in genital IFN-α2a across the study visits in *S. mansoni* PCR/serology+ participants (*n* = 22). Red lines depict mean cytokine concentration at each visit. **h–i** Flow cytometric analysis of HIV entry and CD69 and CCR5 expression in blood CD4+ T cells ex vivo stimulated by IFN-α2a (100 ng/ml). Representative plots for HIV entry, CD69 and CCR5 expression are shown in (**h**) and Supplementary Fig. 9c, d. Experiments were performed with three technical replicates per participant and each data point in (**i**) represents an average of three values per individual. Significance was assessed by paired *t*-test (**a–g**) or Wilcoxon signed rank test (**i**). Source data are provided as a Source Data file. *Sm: Schistosoma mansoni*

signaling was identified among the top 10 enriched pathways (Fig. 4b) and there was post-treatment up-regulation of genes involved in antiviral immunity (Fig. 4a, b, Supplementary Fig. 10), including signal transducer and activator of transcription (*STAT*)-*1*, *IL-22*, *IL-24* and interferon-induced transmembrane protein (*IFITM*)−*1*, a cell membrane-associated inhibitor of

CCR5-tropic HIV entry[33]. Using the interferome database we then assessed whether the DEGs associated with *Sm* treatment were enriched for IFN and IFN-I regulated genes (IRG and IRG-I, respectively). One-month post-treatment DEGs were enriched for both IRG (28%, *p* < 0.001) and IRG-I (17%, *p* = 0.013), while the two-month visit was not enriched for either IRG (24%, *p* = 0.579)

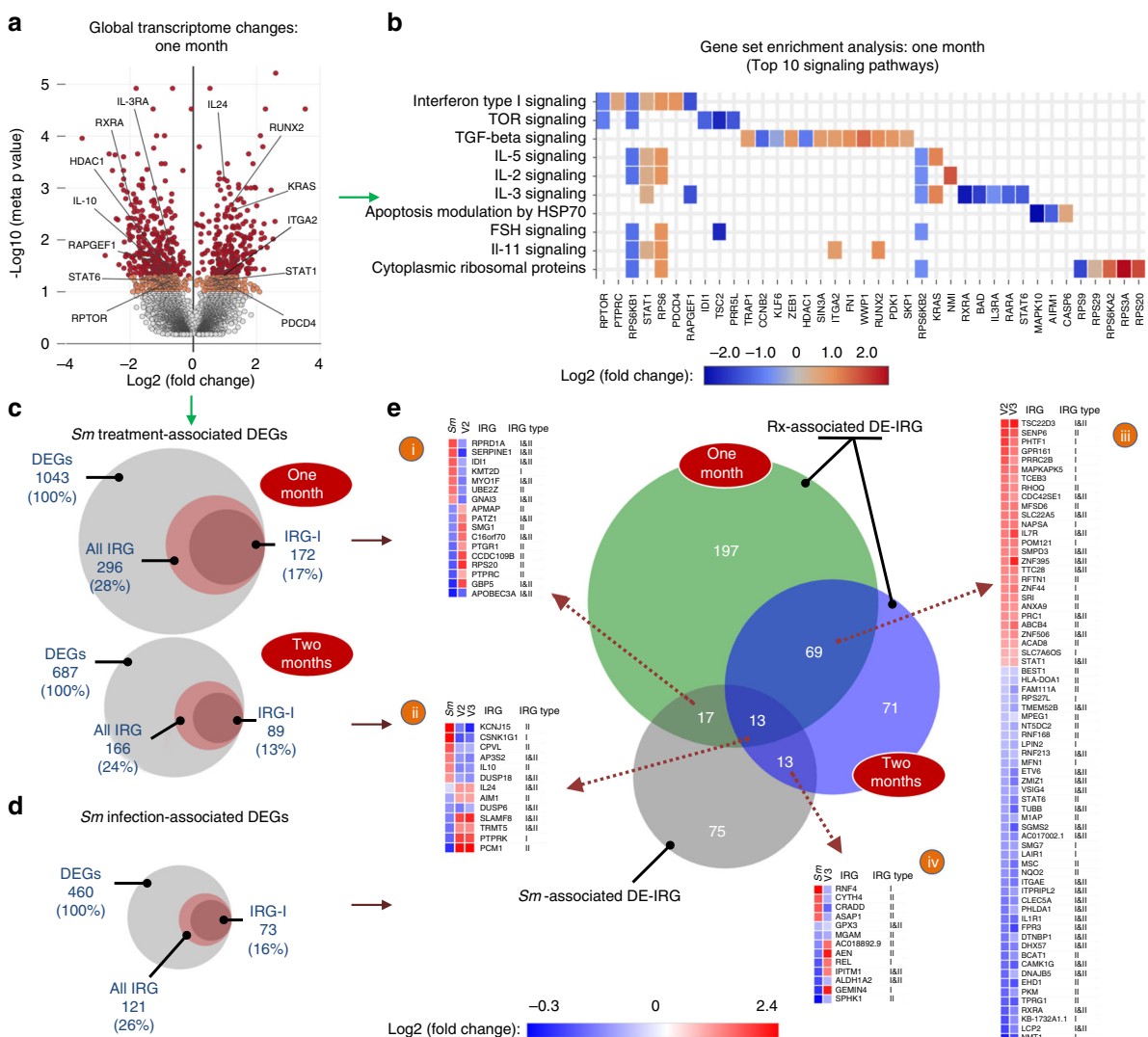

**Fig. 4** Systemic transcriptome and IFN-I signaling changes associated with *Sm* treatment. **a** Global distribution of fold changes and p-values derived from the analysis of genes differentially expressed at one-month post-*Sm* treatment. Red and orange dots denote genes with meta-analysis *p* values ≤ 0.05 and 0.05 < *p* ≤ 0.1, respectively. Cumulatively, out of *n* = 8459 genes, 12.3% (*n* = 1043) were differentially expressed (red and orange), and of the latter 559 (54.0%) were down- and 484 (46.0%) up-regulated. Labels on the volcano plot denote select genes appearing in the gene set enrichment analysis (Fig. 4b); RPTOR (regulatory-associated protein of mTOR), STAT (signal transducer and activator of transcription) 1 and 6, RAPGEF (rap guanine nucleotide exchange factor) 1, HDAC (histone deacetylase) 1, RXRA (retinoid X receptor α), IL-3RA (interleukin 3 receptor α), RUNX (runt-related transcription factor) 2, KRAS (Kirsten rat sarcoma), ITGA (integrin α) 2, PDCD (Programmed Cell Death) 4, and those discussed in the text; IL (interleukin) −10 and IL-24. **b** The top 10 enriched pathways identified by the gene set enrichment analysis using all differentially expressed genes (DEG) identified at the one month (V2) post-treatment time point. TOR (target of rapamycin), TGF (transforming growth factor), IL (interleukin), HSP (heat shock protein), FSH (follicle stimulating hormone). **c** Proportions of IFN and IFN-I regulated genes ("all IRG" and "IRG-I", respectively) differentially expressed at one and two month(s) post-*Sm* treatment. **d** Proportions of all IRG and IRG-I differentially expressed in *Sm*-infected individuals compared to *Sm*-free peers. In **c** and **d**, numbers in brackets denote percentages of each subset relative to the DEG population in each dataset. **e** Venn diagram depicting the overlap between the IRGs differentially expressed after *Sm* treatment (Rx-associated DE-IRG) and the IRGs dysregulated by chronic *Sm* infection (*Sm*-associated DE-IRG). Note that the IRG overlapping between post-treatment visits exhibit the same, while those overlapping between post-treatment visits and prevalent infection show largely the opposite direction of differential expression. See the Source Data file for the complete gene lists and links to interactive plots. *Sm*: *Schistosoma mansoni*

or IRG-I (13%, *p* = 0.747) (Fig. 4c, Supplementary Table 13 and Supplementary Fig. 11), consistent with early elevation of IFN-α2a after schistosomiasis therapy (Fig. 3g). Additionally, in silico cell type enrichment analysis indicated that T cells were the primary cell type influenced by treatment (Supplementary Fig. 12). Overall, the transcriptomic analysis highlighted the immune pathways previously reported to be altered by schistosomiasis therapy (for details see Supplementary Note 5).

**Sm treatment reversed IFN pathway dysregulation**. Having identified IFN-I signaling as a major pathway induced early after *Sm* treatment, we next assessed the impact of *Sm* infection itself on this pathway in a cross-sectional analysis, by performing RNA-seq on PBMC from three additional Ugandan participants who had tested *Sm*- negative by all diagnostic tests and resembled the *Sm* + women chosen for the prospective RNA-seq analysis based on age and socio-behavioral factors (Supplementary

Table 14). Approximately a third (33.7%) of DEGs associated with untreated *Sm* (Supplementary Fig. 13a) overlapped with treatment-associated genes (Supplementary Fig. 14). Notably, the enrichment analysis identified IFN-I signaling among the top 10 pathways associated with prevalent infection (Supplementary Fig. 13b) and there was a trend to IRG-I enrichment of *Sm* infection-associated DEGs (16%, $p = 0.1$) (Fig. 4d and Supplementary Fig. 11). Additionally, *Sm* infection down-regulated several IFN-stimulated genes, including *IFITM1*, *IFITM2*, tripartite motif-containing protein (*TRIM)5*, IFN-α-inducible protein (*IFI)6*, *REL* and *IL-24* (Supplementary Fig. 13a), in line with CD4+ T cell IFN signaling suppression in murine schistosomiasis[4]. We also observed a 9.5% overlap (43/455) between the number of IRGs induced by treatment and those associated with prevalent *Sm* (Fig. 4e) and the IRGs tended to exhibit a reverse direction of differential expression post-treatment compared to prevalent infection (Fig. 4e, panels i-ii and iv). Overall, prevalent *Sm* infection was associated with down-regulation of metabolic and immune signaling (for details see Supplementary Note 6), and *Sm* treatment not only resulted in transient immune activation, but also partly reversed schistosomiasis-associated dysregulation of interferon pathways.

## Discussion

The present study demonstrated that standard praziquantel treatment of *Sm*-infected HIV-negative women resulted in over two- fold reduction of ex vivo HIV entry into cervical- and blood-derived CD4+ T cells. This reduction in HIV susceptibility was accompanied by transient immune activation and induction of antiviral IFN-I signaling, known for its ability to incapacitate multiple stages of HIV infection, from cellular entry to production of virus progeny[33,37]. Remarkably, untreated *Sm* infection was associated with antiviral gene down-regulation, which was partially reversed by praziquantel therapy.

Our results extend findings from earlier studies suggesting that abundant egg destruction after schistosomiasis treatment[38,39] could lead to a surge of IFN-I activity, resulting in acute induction of antiviral defense mechanisms. Induction by *Sm* egg antigens of antiviral and proinflammatory responses was first demonstrated in murine *Sm* granuloma supernatants, which suppressed vesicular stomatitis virus infection via IFN-I and IL-1 activity[40]. It was later confirmed that *Sm* egg antigens such as RNase omega-1[27] and double stranded RNAs[28] activate toll-like receptor 3 and induce an IFN-I signature in conventional dendritic cells[29], an event shown to be necessary for Th2 signaling initiation[27]. Future studies will need to better delineate the dynamics of IFN-I induction by *Sm* eggs in the context of *Sm* treatment and its effects on antiviral immunity.

*Sm* egg-positive study participants had elevated levels of mucosa-homing α4β7 + CD4+ T cells in blood, in keeping with our hypothesis that the inflammation caused by eggs in the gut activates α4β7-mediated common mucosal homing pathways. Sivro and colleagues recently demonstrated that α4β7 expression is linked to elevated HIV susceptibility, and that in HIV-infected people α4β7 is a predictor of higher viral load and faster disease progression[15]. Therefore, our finding of high α4β7 expression in *Sm* + adults suggests a mechanism by which *Sm* could cause not only increased HIV susceptibility, but also lead to faster CD4+ T cell depletion and higher viral loads in HIV co-infected individuals, in keeping with the recent observation of elevated blood HIV RNA levels in *Sm*-infected HIV seroconverters compared to their schistosoma-free peers[10]. Although a one-time praziquantel treatment in our study did not alter the elevated levels of α4β7, the transcriptome analysis indicated that *Sm* treatment was associated with changes in expression of multiple other integrins

by blood mononuclear cells, including α2, αE and αM (see Supplementary Note 5), suggesting treatment-induced reprogramming of cell homing in agreement with studies in murine models of schistosomiasis[41]. Given that schistosomiasis-associated pathological changes are not reversed up to six months post-treatment[42] and eggs retained in tissues continue to exert negative immune effects after chemotherapy[43], it is perhaps unsurprising that we saw only partial restoration of immune function in our study. Future research will need to assess these parameters at later time points to understand the longer-term effects of schistosomiasis therapy on HIV susceptibility and antiviral defenses.

This study, to the best of our knowledge, is the first to document large-scale transcriptomic changes occurring in human PBMC as a result of *Sm* treatment. In keeping with our findings, recent studies identified distinct transcriptional profiles associated with *Sh* infection in cross-sectional comparisons of whole blood- and cervical cytobrush-derived transcriptomes of individuals with and without schistosomiasis[44,45], although the same authors found cervical gene expression to be unaltered by *Sm* infection[45]. Taken together with our results, these data suggest that schistosome infections and their treatment can exert substantial impact on both systemic and mucosal immunity, although the effect size may vary depending on the parasite species and study design.

We found that the follicle stimulating hormone (FSH) pathway was one of the top 10 pathways induced at one month after *Sm* treatment, although this pathway has previously been characterized in reproductive tissues rather than blood-derived lymphocyte populations. This is unlikely related to differences in the menstrual cycle stage between baseline and follow up visits, since we carefully scheduled all study visits to occur at the same stage of the menstrual cycle. While we cannot definitively rule out a direct effect of *Sm* treatment on FSH levels, this pathway shares several of its signaling components with other pathways, and in particular with the mammalian target of rapamycin (mTOR) pathway[46]. In the current study, *Sm* treatment was also associated with down-regulation of the mTOR pathway (Fig. 4b), in line with the role of TOR in the maintenance of T regulatory cells[47] and the demonstrated reduction of these cells post-schistosomiasis therapy[48]. Therefore, it is more plausible that the FSH signature is a result of the overlap of this pathway with other pathways induced by *Sm* treatment, rather than directly reflecting systemic FSH fluctuation, although we were unable to directly measure FSH levels to assess this possibility. Interestingly, our transcriptome analysis also found that untreated *Sm* infection was associated with changes in the sphingosine-1P receptor-mediated signal transduction and down-regulation of CD69 (see Supplementary Note 6), potentially linking the dramatic treatment-induced increase in CD69+ CD4+ T cell frequency to a lift of suppression of immune cell retention and/or tissue residency[31].

While an important limitation of our study is its relatively small sample size, this limitation was overcome by the longitudinal design of the clinical trial, which increased our power to detect differences in immunological readouts by reducing the impact of inter-individual variability. Of all CCA + women in our study, only 35% were Kato-Katz+ at baseline. Although this frequency of *Sm* egg+ individuals may seem low for a *Sm*-endemic region, the low sensitivity of microscopy testing has been documented in women[49] and so was not unexpected. Due to ethical considerations we were unable to perform a larger study of direct praziquantel effects on ex vivo HIV susceptibility in schistosomiasis-uninfected individuals, but the viral entry, CD69 expression and cytokine data obtained from four schistosomiasis-free volunteers suggest it is very unlikely that praziquantel directly mediated the changes we observed in HIV susceptibility and host immunology. While the cost of RNA-seq analysis

limited the sample size of our transcriptome analysis, our paired intra-individual analysis approach provided sufficient power to test the hypothesis regarding IFN-I pathway induction using a meta-analysis approach for DEG selection in combination with gene set enrichment and IRG permutation analyses.

In summary, we found that praziquantel treatment of *Sm*-infected Ugandan adult women substantially reduced ex vivo HIV entry into both endocervical and blood CD4+ T cells for at least two months, despite transient mucosal and systemic immune activation. This reduced HIV entry was associated with elevated mucosal IFN-I levels, and transcriptomic analysis confirmed that *Sm* treatment induced IFN-I pathways and partly reversed expression of *Sm*-dysregulated IRGs, potentially enhancing anti-viral immunity. *Sm* egg shedding was associated with increased expression of a mucosal homing integrin on blood CD4+ T cells, suggesting a mechanism for enhanced genital HIV susceptibility, although α4β7 expression was not substantially reduced within two months of treatment. Identifying mechanisms by which treatment of a neglected parasitic infection could reduce female HIV acquisition is an important step toward designing effective HIV prevention programs. Our findings suggest that mass schistosomiasis treatment may merit investigation as a potential strategy to reduce HIV transmission in co-endemic regions.

## Methods

**Study population and design**. This registered clinical trial (ClinicalTrials.gov #NCT02878564) was conducted in Entebbe, a town situated on a Lake Victoria peninsula with an area of 36.2 km². HIV prevalence in adult Ugandan women from the general population is ~ 7.3%[50], while exceeding 30% in women from lakeshore communities[51]. Schistosomiasis prevalence in lakeshore communities ranges from ~ 50% to over 70% as measured by urine CCA testing[12,17,52]. Schistosomiasis in the region is largely due to *Sm*, with very low (<1%) rates of *Sh*[53]. The study participants did not recall having received any anthelminthic treatment in the last five years preceding the study. The primary endpoint of the clinical trial was the change in the percentage and number of endocervical CD4+ T cells susceptible to HIV pseudovirus entry after treatment of schistosomiasis. Power calculation was performed based on the standard deviation of the difference in viral entry in repeated measures obtained within an individual. Based on these preliminary calculations, recruitment of n = 35 participants would have allowed us to detect a 24% difference in viral entry at β = 0.8.

**Screening procedures**. The initial screening of participants occurred at the Uganda Virus Research Institute (UVRI) community outposts offering free HIV testing and counseling to the communities of lakeshore communities. Consenting HIV-uninfected Ugandan women aged 18–45 years from these communities were tested for schistosomiasis by urine CCA (Rapid Medical Diagnostics, Pretoria, South Africa). All CCA-positive participants were scored by two technologists using a published scoring scheme (Supplementary Table 2)[54] and those scored as " + 1" or above were invited to participate in the study and then screened for inclusion/exclusion criteria. Exclusion criteria were HIV infection, malaria infection, current pregnancy, genital ulceration, active menstruation, positive for classical STIs (*N. gonorrhoeae* (Ng), *C. trachomatis* (Ct), *Treponema pallidum* (syphilis), or *Trichomonas vaginalis* (Tv)- see below for diagnostics), or deemed by study staff to be unlikely to comply with study requirements. Participants, who were not eligible due to the presence of screened infections, were referred to the clinic physician for counseling and treatment.

**Sample collection and schistosomiasis treatment**. Since the study intra-individually assessed immunological changes associated with schistosomiasis treatment, it was important to ensure that samples were collected from the participants at roughly the same point of their menstrual cycle at baseline and follow-up. Therefore, in participants reporting a regular menstrual cycle (21–35 days), the baseline visit was scheduled either in the follicular phase of the menstrual cycle (defined as two days after the last day of the next menstrual bleeding) or the luteal phase (7 days prior to the projected first day of the next bleeding), whichever occurred soonest, and follow-up was scheduled around the same point in their next menstrual cycle. Participants, who were not cycling were followed up 28 ± 2 days after their baseline visit. Questionnaires capturing a range of socio-economic and behavioral characteristics were administered at each visit to the clinic. For studies performed in Canada, healthy blood donor samples were collected from volunteers who were *Sm*-uninfected but taking praziquantel for prevention purposes due to a recent travel history. All participants were provided with a standard dose of 40 mg/kg orally administered praziquantel after collection of biological samples at baseline. The participants were instructed to take praziquantel with their evening meal

to minimize the occurrence of adverse effects associated with drug administration. All participants were provided with detailed instructions on how to take the drug, and under the study protocol the study nurse followed up with every participant on the phone and/or in person to assess compliance and to inquire whether the participant had experienced any compliance difficulties or medication side effects.

**Parasitological assessment**. At each of the three study visits, participants provided urine for CCA testing and scoring done at the UVRI-IAVI clinic on three consecutive days. At the baseline clinic visit stool samples were collected on three consecutive days and transported on ice to the Medical Research Council Clinical Diagnostic Laboratory (MRC-CDL), where parasite egg detection and counting was performed by the Kato-Katz smear technique. An average of three-day CCA scores and egg counts was used in all analyses. Cell free circulating DNA was extracted from stored plasma at the University of Toronto (UofT) using QIAamp MinElute extraction kit (Qiagen, Germany). Subsequently the extracted DNA was shipped to Hamburg, Germany, and *Schistosoma* spp.-specific PCR was performed there as previously described and with Ct values < 45 considered positive[55,56]. *Sm* soluble egg antigen (SEA) serology testing was performed at Public Health Ontario laboratories (Toronto) on stored plasma using a commercially available ELISA kit (Scimedx, USA). Serological optic density (OD) values > 0.2 were considered positive when defining the categorical presence/absence of *Sm/Sh* infection, while raw OD values were used to assess schistosomiasis treatment-associated changes in *Sm*-specific antibody titers. Urine microscopy was performed on 10 ml of urine by standard sedimentation technique to screen for *Sh* eggs at MRC-CDL.

**Genital and blood sampling and testing**. Genital samples were collected at the UVRI-IAVI clinic in the following order: cervico-vaginal secretions, vaginal swabs, and two endocervical cytobrushes. All samples were processed within four hours of collection. A SoftCup (EvoFem, USA) was used to collect cervico-vaginal secretions for mucosal cytokine assays and prostate-specific antigen testing to assess recent coitus (PSA; Seratec PSA Semiquant kit, Göttingen, Germany) and were stored at −80C prior to analyses. One vaginal swab was tested for *Tv* using the OSOM rapid test (Sekisui Diagnostics, USA), and a second vaginal swab was smeared onto a glass slide, air-dried and Gram's stained to diagnose bacterial vaginosis (BV) using Nugent criteria. Endocervical cytobrush was inserted into the cervical os, rotated through 360˚, and stored in R10 medium at 4 °C until processing. Cells from the two cytobrushes were eluted, combined, passed through a 100-μm filter, washed, and divided into two equal aliquots for use in the flow cytometry and virus entry assays at the UVRI-IAVI laboratory. The median numbers of live CD4+ T cells collected per cytobrush were 464 (IQR = 63–1898), 461 (IQR = 113–1146) and 663 (IQR = 323–1956) at V1, V2 and V3, respectively (Supplementary Table 9). Blood was collected by venipuncture into ACD (16 ml) and EDTA (4 ml) vacutainers (BD). Syphilis was tested using SD Bioline syphilis 3.0 (Standard diagnostics Inc.). Full blood counts were acquired using an ACT 5diff automated hematology analyzer (Beckman Coulter, USA) from EDTA blood at the UVRI-IAVI laboratory. Peripheral blood mononuclear cells (PBMC) were isolated from ACD blood by layering onto Ficoll Histopaque (Sigma) and centrifuging at 400 g for 30 min followed by reconstituting at 10 million cells/ml in Roswell Park Memorial Institute (RPMI) 1640 medium (Sigma) with 10% heat-inactivated fetal bovine serum (FBS) (Wisent Inc., Canada). Approximately two million cells were used in flow cytometry assays, the remaining cells were cryopreserved within 6 h of blood draw in 90% FBS and 10% dimethyl sulfoxide in pre-cooled freezing containers (Mr. Frosty, Nalgene) and stored at −150C. Plasma was aliquoted and stored at −80C. Stored plasma was used to perform serology for herpes simplex virus type 2 (Kalon HSV-2 IgG, Kalon Biological Ltd, UK) at UofT. Urine was tested for *Ct* and *Ng* using the Roche Cobas PCR (Roche Diagnostics Corp, USA) at MRC-CDL.

**Production of HIV pseudovirus and the HIV entry assay**. Entry of HIV into CD4+ T cells was measured using a previously described assay[57] based on a β-lactamase containing CCR5-tropic HIV pseudovirus and adapted for use on fresh patient samples[16]. Briefly, cervical and blood mononuclear cells were incubated with the virus overnight to allow viral fusion with target cells and subsequent intracellular release of β-lactamase, which cleaves the FRET acceptor (fluorescein) from the CCF2 dye (commercially available from Invitrogen/Thermo Fisher), producing an emission shift analyzed by flow cytometry on antibody-stained and fixed cells. Pseudovirus stocks were generated at UofT by co-transfecting HEK-293T cells with an HIV backbone lacking envelope (Q23Δenv gfp nef, 20 μg)[58], early-transmitted CCR5-tropic Clade A envelope (Q259d2.17env, 10 μg), cloned from an HIV isolate obtained from a Kenyan woman, 1 week after HIV sero-conversion[59], a plasmid expressing vpr fused to β-lactamase (pCMV-BlaM-Vpr, 10 μg; Addgene, USA), pAdvantage (5 μg; Promega, USA) and 135 μl of the transfection reagent polyethyl-imine (Polysciences, USA). Cell culture supernatant was collected 48 h post-transfection, filtered through a 0.45μm filter and concentrated 100x using PEG6000[60]. Viral stocks were titrated on reference PBMC obtained from HIV-negative donors and quantified by p24 ELISA (Zeptometryx, USA). The quantity of virus used in the study was equivalent to approximately 60% of maximum viral entry in the reference PBMC or 175 ng of p24. The virus stocks were shipped to the UVRI-IAVI laboratory and kept at −80C throughout the duration of the study. The quality control of virus infectivity was performed by

including reference PBMC in infection assays. Processed cervical cell suspensions from each participant were divided in two and spinoculated with either virus (infection well) or media (control well) at 1200 g for 2 h at 17 °C in 48 well flat-bottom plates, followed by a 2-hour incubation at 37 °C, 5% CO2. For PBMC infections, 1 million cells (counted via trypan blue exclusion) were used per well and infected on the same plate. Samples were washed twice in CO2-independent media (CID, Invitrogen/Thermo Fisher) and loaded with 1 μM CCF2-AM (Invitrogen/Thermo Fisher) for 1.5 h. After wash, samples were incubated for 12 h at room temperature in CID supplemented with 10% FBS, antibiotic cocktail (clindamycin, streptomycin, polymyxin, amphotericin, gentamycin) and 250 mM probenecid (Invitrogen/Thermo Fisher). Samples were then stained with fluorescently-labeled antibodies and a live-dead dye (see Supplementary Table 7 for details) and analyzed using flow cytometry. HIV entry into CD4+ T cells was analyzed by assessing cleaved/uncleaved CCF2 ratio and using the corresponding uninfected well to guide the gating (Fig. 2a and Supplementary Figures 3–4).

**Flow cytometry**. Flow cytometry was performed on either BD LSR-II or BD LSR Fortessa X-20 (BD Biosciences) cytometers at the UVRI-IAVI laboratory or UofT, respectively. Compensation was performed regularly using single stained compensation beads (BD Biosciences). The fluorescence output of the cytometer channels across the runs was standardized using Ultra rainbow calibration particles (Spherotech, USA). Analysis was performed using FlowJo software v.10.4.1 (TreeStar) with blinded participant/study visit identifications. Gating was guided by fluorescence minus 0 (FMO) controls. Cervical samples were excluded from analysis if the CD4+ T cell count per FCS file was less than 15 cells and/or if the control well appeared contaminated by pseudovirus.

**Multiplex ELISA on plasma and genital secretions**. Plasma was collected after PBMC separation and centrifuged at 1000 g for 10 min prior to freezing. Soft cup secretions were spun down at 500 g for 5 min, re-suspended in 10x original mass in 1X Dulbecco's phosphate buffered saline (Gibco/Thermo Fisher) and centrifuged for 10 min at 1000 g. All cytokine assays were performed at UofT. Thawed plasma and genital secretions were tested by Meso Scale Discovery V-PLEX ELISA using pre-made (Cat. # K15049D) and customized panels on Sector Imager 6000 (Meso Scale Diagnostics, USA) for most cytokines. Human panIFN-alpha ELISA kit (Stemcell technologies, Canada) was used to assess the levels of pan IFN-α (specific but unable to distinguish IFN-α sub-types 1/13, 2, 4–8, 10, 14, 16 and 17). To avoid the bias of intra-assay and inter-plate variation, paired samples were examined in duplicate on the same plate. Re-testing in a paired fashion was performed if one or more samples exhibited a coefficient of variation >30%. Standard curves were produced according to the manufacturer's instructions. The quantitative characteristics for the assays are listed in Supplementary Table 10. Cytokine heatmaps were constructed by first calculating "follow-up/baseline" ratios of cytokine levels for each participant, then a geometric mean of individual ratios was calculated for each cytokine and graphed using Morpheus (https://software. broadinstitute.org/morpheus/).

**Ex vivo effects of IFN-α on CD4+ T cells and HIV entry**. Blood was drawn from schistosomiasis-free Canadian volunteers (n = 5): these were four women and one man with a median age 22 (IQR = 21.5–28) years and without a recent history of travel to schistosomiasis-endemic regions. PBMC were isolated as described above. Cells were plated into 48-well plates and rested for 24 h at 37 C. IFN-α2a (StemCell Technologies, Canada) reconstituted in RPMI was used to stimulate cells for 24 h at 37 C at a final concentration of 10, 50 or 100 ng/ml- the range of IFN-α2a concentrations chosen based on the quantities of IFN-α2a commonly used to induce interferon pathway signaling in a laboratory setting[61,62]. Subsequent to incubation with IFN-α2a, PBMC were infected with Blam-Vpr virus in triplicate and analyzed using flow cytometry at UofT as described above.

**RNA-seq experiments; Cell handling and RNA extraction**. Stored PBMC were thawed in 10% FBS/RPMI 1640 at RT and manually counted via trypan blue exclusion at UofT. Approximately two million cells were used for RNA extraction using RiboPure RNA Purification Kit (AM1924, Ambion/Thermo Fisher) following the manufacturer's instructions. Extracted RNA was treated with DNase using DNA-free DNA Removal kit (AM1906, Invitrogen/Thermo Fisher) prior to RNA quality and quantity assessment with an Agilent Bioanalyzer (Agilent) and Qubit RNA kit (Life Technologies/Thermo Fisher), respectively.

**RNA library preparation and sequencing**. Samples selected for library preparation had an RNA integrity score >7.0. Libraries were prepared using TruSeq Stranded Total RNA kit (Illumina Technologies). A starting amount of 100 ng of RNA for each sample was depleted of cytoplasmic and mitochondrial ribosomal RNA using Ribo-Zero Gold rRNA beads (Illumina Technologies), followed by RNA fragmentation. The cleaved RNA fragments were copied into first strand cDNA using reverse transcriptase and random primers, followed by second strand cDNA synthesis using RNase H and DNA Polymerase I. A single "A" base was added prior to adapter ligation and then following purification cDNA library enrichment was performed using PCR. The final cDNA libraries were validated for size and concentration using Agilent Bioanalyzer and qPCR (Kapa Biosystems/

Roche), respectively. All libraries were normalized to 10 nM and pooled together, denatured with 0.2 N NaOH and diluted to a final concentration of 1.4 pM. The pooled libraries (1.3 ml of 1.4 pM) were loaded onto an Illumina NextSeq cartridge for subsequent cluster generation and sequencing on an Illumina Nextseq 500 instrument (Illumina Technologies) at Princess Margaret Genomics Centre (www. pmgenomics.ca) using the paired-end 75 bp protocol to achieve ~ 30 million reads per sample.

**Transcript alignment and assembly**. Overall read quality was checked using FASTQC v.0.11.5. The raw sequence data, in the form of FASTQ files, was aligned to the human genome (GRCh38, Ensembl Homo_sapiens.GRCh38.84.gtf definition file) using HISAT2 (v. 2.1.0) and SAMTOOLS (v1.3.1). Transcript assembly was done using StringTie (v1.3.3b) and read count for each sample was generated with HTSeq (v0.7.2)[63].

**Differential expression analysis**. Analysis of *Sm* treatment-associated changes consisted of a paired comparison of gene expression profiles at one and two months post-treatment vs. the baseline (pre-treatment) profile. Our cross-sectional analysis of prevalent *Sm* consisted of an unpaired analysis of individuals with and without confirmed *Sm* infection. The analysis had two goals: (i) to perform gene set enrichment analysis on the treatment- and schistosomiasis- associated DEGs and (ii) to assess IRG enrichment of the identified DEGs. HTSeq counts were uploaded into RNA-seq 2 G[64] (http://52.90.192.24:3838/rnaseq2g/) for analysis. For the paired prospective study analysis, the "paired" test was specified, while the unpaired test was specified for the cross-sectional comparison of "schisto+ " vs. "healthy controls". Normalization was performed using default settings ("normalize count by DESeq"/"normalize logged by Loess"). Minimal read count threshold was set to "one". To increase the power and reduce the uncertainty of our DE comparison, we performed meta-analysis[65] using five DE pipelines: DESeq2, edgeR, limma, PoissonSeq and Ballgown (all implemented using metaseqR in RNA-seq 2 G[64]). These pipelines were chosen based on 1) their overall recognition/ prior use in the field (DESeq2, edgeR, limma, Ballgown), 2) capacity to perform paired analysis (all) and 3) bias to identify genes based on length (PoissonSeq being the least biased of all[64]). Meta-analysis was performed for each comparison (V1 vs. V2, V1 vs. V3, schisto+ vs. schisto-,) using the RNA-seq 2 G default settings (combined p value estimated by the Simes method, p values normalized). The distribution of meta-analysis p values and fold changes were plotted using Plot.ly (https://plot.ly/) (Fig. 4a, Supplementary Figs. 10a, 13a). Since the goal of our analysis was to test a specific hypothesis about the IFN-I pathway induction (but not to characterize individual DEGs), we considered a liberal sample of genes that had p values of 0.1 or lower, regardless of the fold change, derived from the meta-analyses for querying against the Enrichr and Interferome databases.

**Gene set enrichment analysis**. We used WikiPathways[66] via Enrichr[67], both freely available and regularly updated biological pathway databases. Lists of DEGs were submitted to Enrichr without pre-specified levels of gene membership. The top ten pathways identified by Enrichr/WikiPathways were plotted in Plot.ly including their member genes and fold changes (Fig. 4b, Figs S10b, S13b).

**IRG analysis**. The DEGs from the RNA-seq analysis were queried against the Interferome database (v. 2.01)[68] by specifying the organism as "human", system as "hemopoietic/immune" and organ as "blood". The output was then cleaned to remove duplicates and sorted by IRG type. At the time of analysis (September 2018), the human Interferome contained 7350 and 9768 IRG-I and IRG-II, respectively, with experimental p-values < 0.05 and fold changes ≥ 2.0, while the IRG-III subset was under-represented (184/12,614 genes) and therefore excluded from our analysis.

**Permutation analysis to assess IRG enrichment**. A one sample proportion test was performed to assess whether the proportion of IRG/IRG-I genes in each dataset was the same as that expected from random sampling from the complete dataset of all (differentially and non-differentially) expressed genes. To this end, we took all the genes (n = 8459) expressed in our RNAseq dataset and queried these genes against the Interferome database to obtain the proportions of IRG and IRG-I subsets (2082/8459 and 1102/8459, Supplementary Table 13) in the complete dataset. We then used these proportions to estimate the number of IRG/ IRG-I subsets expected at post-treatment visits (V2 and V3) and in the cross-sectional analysis by creating random samples of size n (n[V2] = 1044, n[V3] = 688, n[Xsec] = 461, Supplementary Table 13), in which the expected IRG and IRG-I frequencies were derived by drawing from a binomial distribution with a probability equal to the expected proportion of IRG and IRG-I subsets. The random samples were then shuffled 2000 times to create an approximate permutation null distribution and compared to the observed number of IRG and IRG-I genes in each dataset. The statistical significance of the observed number of genes was assessed by calculating an approximate empirical p-value from the null distribution (Supplementary Fig. 11).Venn diagrams and gene expression heatmaps were plotted using BioVenn[69] and Morpheus (https://software.broadinstitute.org/morpheus/), respectively.

**Cell type enrichment**. Raw RPKM counts were submitted to the xCell[70] server (v1.1) using default settings (xCell gene signature, n = 64). The xCell enrichment scores are estimates of the relative contribution of each cell subtype to the global transcriptome. The difference in enrichment for major lymphocyte subtypes and monocytes was compared for each participant (Supplementary Fig. 12).

**Statistical analysis**. Statistical analyses were performed using IBM SPSS v.25 and graphing using GraphPad Prism v.7, unless otherwise specified. The data were checked to confirm residual normality, and two-tailed parametric significance testing was used for data that fit the criteria for normality, otherwise two-tailed non-parametric tests were used as specified. Where necessary, log10 transformation was applied to scale data. The flow cytometry, viral assay and individual log-transformed cytokine data were analyzed using paired t-test or repeated measures ANOVA as specified. The viral and flow cytometry assays performed on healthy controls and complete blood count changes were analyzed using Wilcoxon signed rank test. The permutation analysis and histogram plotting were performed using R version 3.5.1 using code available in the Supplementary Information. All fold changes are median fold changes derived from paired intra-individual comparisons, unless specified otherwise.

**Ethics statement**. All study procedures were approved by the Uganda Virus Research Institute Research and Ethics Committee, the Uganda National Council for Science and Technology, and the Institutional Review Board at the University of Toronto. Written informed consent was obtained from all participants.

**Reporting summary**. Further information on research design is available in the Nature Research Reporting Summary linked to this article.

## Data availability

All flow cytometry files were deposited in the FlowRepository database (https://flowrepository.org) under experiment IDs FR-FCM-ZYV5, FR-FCM-ZYVF and FR-FCM-ZYVD. All RNA sequencing files were deposited in the short read sequence archive (http://www.ncbi.nlm.nih.gov/sra) under BioProject ID PRJNA522847. The source data for all figures and tables are included here as a Source Data spreadsheet file. The R code used to perform enrichment analysis is included in the Supplementary Data. The authors declare that all other data supporting the findings of this study are accessible within the article and its Supplementary Information files, or are available from the authors upon request.

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

## Acknowledgements

We thank all the participants, community leaders and research teams involved in the study. We are especially grateful to Dr. George Miiro, Obenyu P. Akiteng and David A. Drajole at the UVRI clinic for help with participant screening and recruitment, Sanja Huibner at UofT for technical assistance with MSD ELISA assays and cell free DNA extractions, Rachel Lau at the PHO Laboratories for technical assistance with serology, Dr. Dominic Wichmann at University Hospital Hamburg-Eppendorf for his expert advice on *Schistosoma* spp. PCR, Dionne White at the UofT Flow Cytometry facility for technical support and Andrew Mubiru for help with the experimental assays. This study was supported by the Canadian Institutes of Health Research (CIHR) grant to RK (TMI-138656). SY was supported by a CIHR Vanier Canada Graduate Scholarship. RMG was supported by the Fogarty HIV Research Training Program of the National Institutes of Health (4D43TW009578-04).

## Author contributions

S.Y., R.K.: conceived, designed and implemented the study, drafted the manuscript. S.Y., V.J. and R.M.G. conducted experiments including production, optimization and quality control of HIV pseudovirus. S.Y., R.M.G.: participated in participant recruitment. S.V.G.: assisted with participant recruitment and RNA-seq experiment design. S.Y., S.V.G., R.K.: analyzed the data. J.M.: contributed to study design and supervision at clinical sites. E.T., A.B.: oversaw schistosomiasis PCR/serology testing. N.K., B.S.B., R.K.: overall study conception, design, implementation and supervision. All authors critically reviewed the manuscript draft and approved the final version of the article.

## Additional information

**Competing interests:** The authors declare no competing interests.

