## [Peer Review File · Nature Communications]

Reviewers' comments:

Reviewer #1 (Remarks to the Author):

This is a novel study that describes efforts to determine the mechanisms that underlie the epidemiological observation that *S. mansoni* infection increases the risk of HIV acquisition in women. The investigators report that, following treatment of *S. mansoni* infection, women had induction of genital and systemic circulation IFN-I pathways, suggesting an increase in anti-viral immunity. They further showed that HIV entry into cervical and systemic CD4+ T cells was reduced by two-fold after treatment of schistosomiasis. They also showed persistent $\alpha 4\beta 7$ homing integrin on blood CD4+ cells that did not change after anti-schistosomiasis treatment.

This paper represents important translational work and is novel in its focus on the genital tract responses to *S. mansoni* infection, which typically affects the gastrointestinal tract. The authors' findings fit with epidemiological observations and they present a large amount of data that encompasses several different means of testing their hypothesis (PBMCs, in vitro experiments, transcriptomic analysis). If strengthened and sufficiently robust, the paper would be of high interest to readers of Nature Communications. As it presently stands, it has some key weaknesses as discussed below.

Major Comments:

1. More detail is needed about the recruitment of the study participants. Why were 9 different villages needed to recruit such a small sample size? How were the women sought for study participation – are they representative of the general population, or were they recruited from a certain clinic or were seeking care for symptoms? It is also not clear from the way that this is written whether this study fits into a larger clinical trial and these patients are a subset from that trial, or whether this study is itself the registered clinical trial.
2. The recruitment of women at different phases of their menstrual cycles may have significantly affected the study's findings. (See, for example, PMID 15126584 that shows lower interferon responses during the luteal phase or PMID 27170437 showing many differentially expressed genes from women in the luteal versus follicular phases of their cycles). There is also no mention of whether women were at the same point in their menstrual cycle at the follow-up visits. If many women were in the luteal phase at baseline (as shown in Table S1) and then they were in the follicular phase at follow-up, this could falsely give the appearance of increased interferon responses at follow-up. It seems that this could have been the case particularly given that the FSH pathway (which I assume stands for Follicular Stimulating Hormone) is one of the 10 differentially expressed pathways from baseline to follow-up.
3. Who were the four people who are later mentioned who were negative for schistosomiasis and treated with praziquantel and how do they fit into this study population? Were they all women like the rest of the 34 participants? Why were they taking praziquantel for "prevention"? This would imply that they may have traveled to an endemic setting and therefore would need to have been screened for schistosomiasis similarly to the Ugandan women? Do the authors have samples from these people (or ideally a larger group of women negative for schistosomiasis), and could the authors compare the same characteristics of women with and without *S. mansoni* infection as those presented in this paper for women pre- and post-treatment? The authors' findings would be further strengthened if they could report that at baseline they saw lower levels of IFN-I in the women with *S. mansoni* infection than in uninfected women, and then that these levels increased following praziquantel treatment.
4. Figure 1 – from the text it seems that confirmation of *S. mansoni* was done by PCR but from the figure it seems that either Sm specific antibodies or Sm specific PCR was included. Please clarify which and how many were discordant by these two techniques. Aside from this, Figure 1 is very clear and helpful.
5. How was praziquantel administered? Was medication use confirmed/observed? Presenting the CCA data as the median decrease is a bit obscure. How many women became CCA negative? Were these treatment results confirmed by PCR? It seems that presenting the actual results more clearly

would be helpful.

6. Calling the women who were excreting any *S. mansoni* eggs in stool “high burden” is confusing. Typically studies report “high burden” as the women excreting a large number of eggs per gram of stool. Rather, the authors’ finding of 12 out of 34 women excreting eggs is likely due to the reported lower sensitivity of Kato Katz testing in women with schistosomiasis (see PMID 29405114).

7. The authors mention that the prevalence of *S. mansoni* is 70% and of HIV is 20%, but the numbers observed in Figure 1 do not fit with this. Is there a reason? What treatment has taken place in these villages recently and do the authors have data on how recently people report being treated? This is important given the findings in references 9-10 cited by the authors in which a history of schistosomiasis treatment was associated with increased risk of HIV acquisition.

8. The authors should be commended for including a sensitivity analysis in those with confirmed *S. mansoni* infection. Could they perform a similar sensitivity analysis leaving out women with BV?

9. Supplemental Tables S6 and Figure S5. Why does one of these use the *S. mansoni* confirmed group as a comparator and the other use the Kato Katz confirmed group as a comparator?

10. Did (or could) the authors assess the frequency of a4b7 cells in the cervical mucosa? In concordance with their hypothesis that circulating systemic a4b7 cells would be elevated and would home to the genital mucosa, these would be expected to have been elevated as well.

11. Lines 176-8 – Why were only 3 women included in the RNA-Seq analysis when there were 12 who would have been eligible (if the criteria was CCA+KatoKatz+)? Was this subset pre-specified? Were these people who cleared their *S. mansoni* infection at V2 and V3? Did they have any co-infections? More detail is needed.

12. Lines 350-1 – Who are the schistosomiasis-free volunteers used for the ex vivo CD4+ T cell phenotype and HIV entry study, and how were they chosen? Were they also the Canadian volunteers receiving praziquantel for prevention?

13. Please provide better justification for including all genes that have a $p < 0.10$ rather than $p < 0.05$ in your analysis. Is this something routinely done by other investigators?

Minor comments:

Line 63 – the reference for the prevalence of *S. mansoni* infection is 15 years old – please use a more updated source.

Line 67 – I suggest being more precise with language. *S. haematobium* eggs are not “secreted” into the lumen of the gut; rather, they migrate there.

Line 87 – The fact that HIV positivity was an exclusion criterion is important, since HIV infection could alter immune responses in women, so should also be mentioned early in the text (not just in the methods and Figure 1).

Lines 245 – 247 – Please provide justification for including women in either the follicular or luteal phases of their menstrual cycles (rather than at a single time point in the menstrual cycle).

Lines 268-269 – More details on the PCR used for *S. mansoni* is needed – such as primers, reaction, etc and/or “as previously described.”

Lines 283-285 – More details on the cervical cell collection and preservation would be useful. Where was the processing performed? What percent viability was obtained? How many cells were obtained per cytobrush? Throughout the methods, please clarify where the testing was performed – at University of Toronto or UVRI or elsewhere.

Figure 4 presents a large amount of data and the abbreviations for at least some things should be shown clearly – for example, at least the names of the top 10 signaling pathways and the genes

pointed out on Fig 4a. Also, it would seem that any genes pointed out on Fig 4a should also be seen on Fig 4b.

Supplemental Table 2 – It would be useful to include the numbers of women that were CCA 1+, 2+, and 3+ and the median number of eggs per gram detected with Kato Katz screening.

Supplemental Figure 3 – Please show the numbers of patients included in each group and mention again here how you defined those with “high *S. mansoni* burden.”

Supplemental Figure S8 – The figure is illegible.

Reviewer #3 (Remarks to the Author):

This is a very interesting manuscript that provides convincing mechanistic evidence for how *Schistosoma mansoni* infections may increase a woman’s susceptibility to HIV. This is a theory that has been around for over 20 years and there are studies that both support and contradict the idea. However, the majority of the work has been performed by schistosomiasis researchers with occasional support from HIV investigators. Therefore, addressing the question from an HIV perspective and using sophisticated techniques (e.g. the pseudovirus entry assay) not typically familiar to parasitologists is a welcome contribution. For the same reason, the limitations of the paper have to do with the authors’ more limited understanding of schistosomiasis. However, these limitations are by no means deal breakers and can be easily addressed with some rewriting.

The primary example of this is the way the authors refer to the various infection groups. “High *S. mansoni* burden” has a very specific definition by the WHO: >400 eggs per gram of stool. While there may be 2-3 people in the study that meet this definition, using it to designate all those who were egg positive is inappropriate. Rather than using “whole cohort”, “confirmed *S. mansoni*”, and “high *S. mansoni* burden”, the authors should simply use CCA+, PCR/serology+, and Kato-Katz+. They have separately demonstrated that there are no *S. haematobium*-infected individuals in the cohort so the concept of “*S. mansoni* confirmed” is somewhat redundant. The change in group definitions should be changed throughout the manuscript and supplemental material.

Were PCR and Kato-Katz tests performed at V2 and V3? It should be stated either way and the results should be included (how many previously positive became negative) if these tests were done.

There is a greater impact at V2 than V3 for several of the measures the authors used. Do the authors have any insight on whether the reduction in HIV infections susceptibility and increased IFN production is a transient or more lasting effect? Of course it has more public health impact if it is more permanent. Addressing this question in the discussion (even if the answer is “more research”) would be helpful.

Minor points/wording suggestions:

- 1) Line 51: use “Praziquantel treatment for schistosomiasis” rather than “Schistosomiasis treatment”
- 2) Line 60: add “for women with intestinal schistosomiasis” to the end of the sentences after “susceptibility”
- 3) Line 62: add “in *Sm*-endemic areas” to the end of the last sentence.
- 4) Line 66: change “egg-contaminated water” to “water containing infectious cercariae”
- 5) Line 70: add a sentence to the effect of “In addition, egg deposition induces a strong Th2 response that can cross regulate Th1 responses.”
- 6) Line 87: change “schistosomiasis” to “schistosome” The parasite causes the infection, schistosomiasis is the disease.
- 7) When describing the communities in Table S1, it would be informative to include a column denoting how far each village is from Lake Victoria. Also, what is the overall size of the study area? Alternatively, the authors could include a map of the study villages as a supplemental figure.
- 8) Line 97: replace “were deemed to have a high worm burden based on” with “had”

- 9) Line 101: shouldn't V1 and V2 be V2 and V3?
- 10) Lines 104-106: include PCR and Kato Katz data for follow up time points if you have it.
- 11) Line 143: change "schistosomiasis treatment" to "praziquantel treatment"
- 12) Line 149: replace "a high Sm burden" with "being egg positive"
- 13) Figure 2: panels g and h are not labeled.
- 14) Line 166: change infection designations
- 15) Line 178: egg positive rather than high worm burden. Figure 4a is not mentioned in the text.
- 16) Line 222: Being egg positive rather than High Sm burden
- 17) Line 651: change entire cohort to CCA+ and high worm burden to Kato-Katz+ (please check all text and figures—I have not necessarily noted every place it needs changing).

Reviewer #1 (Remarks to the Author)

This is a novel study that describes efforts to determine the mechanisms that underlie the epidemiological observation that *S. mansoni* infection increases the risk of HIV acquisition in women. The investigators report that, following treatment of *S. mansoni* infection, women had induction of genital and systemic circulation IFN-I pathways, suggesting an increase in anti-viral immunity. They further showed that HIV entry into cervical and systemic CD4+ T cells was reduced by two-fold after treatment of schistosomiasis. They also showed persistent $\alpha 4\beta 7$ homing integrin on blood CD4+ cells that did not change after anti-schistosomiasis treatment.

This paper represents important translational work and is novel in its focus on the genital tract responses to *S. mansoni* infection, which typically affects the gastrointestinal tract. The authors' findings fit with epidemiological observations and they present a large amount of data that encompasses several different means of testing their hypothesis (PBMCs, in vitro experiments, transcriptomic analysis). If strengthened and sufficiently robust, the paper would be of high interest to readers of Nature Communications. As it presently stands, it has some key weaknesses as discussed below.

Authors' response: We are very appreciative of the reviewer's positive view of our work and thank the reviewer for the detailed comments that helped us to substantially improve the manuscript. Please kindly find our point-by-point responses to the reviewer's remarks below.

Major Comments:

1. More detail is needed about the recruitment of the study participants. Why were 9 different villages needed to recruit such a small sample size? How were the women sought for study participation – are they representative of the general population, or were they recruited from a certain clinic or were seeking care for symptoms?

Authors' response: We have now added more detail about the recruitment process in the Methods (pg16, lines 20-24) and Results (pg6). Specifically, participants were recruited through an established network of UVRI-IAVI community outreach sites from the general population of lakeshore communities residing in the radius of approx. 2.8 km from Lake Victoria.

It is also not clear from the way that this is written whether this study fits into a larger clinical trial and these patients are a subset from that trial, or whether this study is itself the registered clinical trial.

Authors' response: We apologize for any lack of clarity. The study is itself an independent registered clinical trial and is not a subset of a larger trial. This is now specified in the Results and Methods sections.

2. The recruitment of women at different phases of their menstrual cycles may have significantly affected the study's findings. (See, for example, PMID 15126584 that shows lower interferon responses during the luteal phase or PMID 27170437 showing many differentially expressed genes from women in the luteal versus follicular phases of their cycles). There is also no mention of whether women were at the same point in their menstrual cycle at the follow-up visits. If many women were in the luteal phase at baseline (as shown in Table S1) and then they were in the follicular phase at follow-up, this could falsely give the appearance of increased interferon responses at follow-up. It seems that this could have been the case particularly given that the FSH pathway (which I assume stands for Follicular Stimulating Hormone) is one of the 10 differentially expressed pathways from baseline to follow-up.

Authors' response: The reviewer is certainly correct that the menstrual cycle has very important effects on genital immunology, and we actively took this into consideration in our study design. Specifically, as we have now clarified in the Methods section (Pg17 line20-Pg18 line 4), post-treatment follow-up of participants with an active menstrual cycle was scheduled to occur at the same point in their cycle (-/+2 days) as their baseline visit; for participants who were not cycling, follow-up was scheduled 28-/+2 days after their baseline visits. This was also the case for the subset of women in whom RNAseq analysis was performed. We have now added information in the Results section regarding the median/IQR of follow-up time post-treatment for both the larger (Results Pg7, lines 6-9) and the RNAseq group (Pg 11, lines 12-13). Indeed, the fact that the FSH pathway appears as one of the 10 top pathways induced by *Sm* treatment emphasizes the importance of the cycle on host immunity, and may reflect that fact that the FSH pathway shares several proteins with other pathways including the mTOR pathway, which was also induced post-Rx and is a signature of regulatory T cells (consistent with downregulation of Tregs post schisto Rx, as also discussed now in the Discussion, Pg 15 lines 3-11).

3. Who were the four people who are later mentioned who were negative for schistosomiasis and treated with praziquantel and how do they fit into this study population? Were they all women like the rest of the 34 participants? Why were they taking praziquantel for "prevention"? This would imply that they may have traveled to an endemic setting and therefore would need to have been screened for schistosomiasis similarly to the Ugandan women?

Authors' response: Thank you for noting this. We have now clarified in the Results section (Pg10 lines 3-8) the nature of this control group, which consisted of three women and one man who received empiric PZQ treatment in Canada due to a recent history of travel in a schistosomiasis-endemic region (also in Methods Pg18 lines 5-7 and in Suppl. Table 11). All participants were deemed uninfected by schistosomiasis on the basis of combined urine CCA, serology and PCR. Practical considerations

precluded a larger sample size: specifically, our REB recommended that investigator initiated off-label use of praziquantel for a research study would require formal registration and monitoring as a nationally-registered clinical trial, something that was beyond our time and budget limitations. This limitation is now discussed in the Discussion (pg15 line 21- pg16 line 1).

Do the authors have samples from these people (or ideally a larger group of women negative for schistosomiasis), and could the authors compare the same characteristics of women with and without *S. mansoni* infection as those presented in this paper for women pre- and post-treatment? The authors' findings would be further strengthened if they could report that at baseline they saw lower levels of IFN- γ in the women with *S. mansoni* infection than in uninfected women, and then that these levels increased following praziquantel treatment.

Authors' response: Unfortunately, we do not have genital samples from these control participants since they were not provided with praziquantel in the context of the same clinical trial protocol.

The reviewer makes an excellent point that our findings might be further strengthened if we also found alterations in IFN- γ levels and/or pathways between Ugandan women with and without schistosomiasis in a cross-sectional study format. However, we did not include such a group in our study for two reasons. Most importantly, we had previously informed trial design by performing a pilot cross-sectional study in women from the region that collected behavioural data and schistosomiasis diagnostics [Ref#52 in the text: Yegorov et al, BMC Infect Dis, 2018]. This study, entitled "*Schistosoma mansoni* infection and sociobehavioural predictors of HIV risk: a cross sectional study in women from Uganda", found that there were important differences between women with and without *S. mansoni* infection that would be expected to confound cross sectional studies of genital immunology: these differences included age, marital status, contraceptive use, time from last sex and chlamydia infection. Therefore, we opted to restrict our studies of genital immunology to a prospective trial design that would be less affected by such potential confounders. A more minor concern was our ability to truly rule out in the field low grade, subclinical *S. mansoni* infection among uninfected controls in such a hyperendemic infection context.

4. Figure 1 – from the text it seems that confirmation of *S. mansoni* was done by PCR but from the figure it seems that either *Sm* specific antibodies or *Sm* specific PCR was included. Please clarify which and how many were discordant by these two techniques. Aside from this, Figure 1 is very clear and helpful.

Authors' response: The reviewer is correct that species-specific diagnosis was based on positive PCR and/or serology. These tests, which are less sensitive than the urine CCA, confirmed *Sm* infection in 24/34 women (71%). Of these 24 participants, 20 were dually positive, 1 was only positive by PCR and 3 were only positive by serology; no cases of *Sh* infection were

detected. These data have now been added to the Results (Pg 6 lines 18-21) and described in detail in Suppl. Table 3.

5. How was praziquantel administered? Was medication use confirmed/observed?

Authors' response: This is now clarified in the Methods (Pg18 lines 7-13). Medication use was not directly observed, largely due to participant desire to take the 6 tablets (an average adult dose) with food (usually their evening meal). All participants were provided with detailed instructions on how to take the drug, and under the Study Protocol the study nurse followed up with every participant on the phone and/or in person to assess compliance and to ask whether the participant had experienced any compliance difficulties or medication side effects. In addition, the analysis of CCA scores before and after praziquantel therapy demonstrated a post-treatment reduction of CCA scores in ~90% of participants (Results pg 7, lines 12-14 and Suppl Tables 4 & 5), providing evidence of good compliance.

Presenting the CCA data as the median decrease is a bit obscure. How many women became CCA negative? Were these treatment results confirmed by PCR? It seems that presenting the actual results more clearly would be helpful.

Authors' response: This has now been clarified in the Results (pg 7, lines 12-14), Methods and in Suppl. Tables 4 & 5. Specifically, PCR was performed at V1 and V3, as described now in the text (Pg 6, lines 17-20), in a majority of participants there was no change of PCR status post-treatment (Pg7, lines 18-20).

6. Calling the women who were excreting any *S. mansoni* eggs in stool "high burden" is confusing. Typically studies report "high burden" as the women excreting a large number of eggs per gram of stool. Rather, the authors' finding of 12 out of 34 women excreting eggs is likely due to the reported lower sensitivity of Kato Katz testing in women with schistosomiasis (see PMID 29405114).

Authors' response: Thank you for noting this. We have now changed this terminology throughout the manuscript, and have added this reference to the Discussion (Pg 15, line 23).

7. The authors mention that the prevalence of *S. mansoni* is 70% and of HIV is 20%, but the numbers observed in Figure 1 do not fit with this. Is there a reason? What treatment has taken place in these villages recently and do the authors have data on how recently people report being treated? This is important given the findings in references 9-10 cited by the authors in which a history of schistosomiasis treatment was associated with increased risk of HIV acquisition.

Authors' response: We apologize for the discrepancy in *Sm* prevalence, which we think is due to heterogeneity in published study results rather

than to any recent community-wide interventions. We have now clarified in the Methods (Pg16 line 22- Pg17 line 2) the prevalence ranges observed in the region for schistosomiasis are 50-70%, meaning that the schistosomiasis prevalence we observed (54%) is in agreement with the literature.

In terms of HIV, we believe that the low prevalence among screened women was due to the fact that our community outreach workers and forms made it clear that only HIV-uninfected individuals were eligible, so that individuals aware of their HIV+ status would be unlikely to come forward.

We have also specified in the Methods that “study participants did not recall having received any anthelmintic treatment in the last 5 years preceding the study” (Pg 17, lines 3-4).

8. The authors should be commended for including a sensitivity analysis in those with confirmed *S. mansoni* infection. Could they perform a similar sensitivity analysis leaving out women with BV?

Authors’ response: This is an excellent suggestion, since BV has been shown by several groups to have important mucosal immune effects. We have performed two sensitivity analyses focusing on the study’s primary endpoint (viral entry at one month post-*Sm* treatment): one excluding participants with BV at visits V1 and V2, and the other excluding participants whose BV status changed between visits V1 and V2. As shown in the table below, our results remained significant in these sub-analyses, and a summary sentence to this effect has been added to the manuscript (Pg 8, lines 12-13).

Table. Changes in HIV entry into cervical and peripheral blood CD4 T cells before (V1) and after schistosomiasis treatment (V2) in participants whose BV status did not change between baseline (V1) and one month post-treatment (V2) and in the subset without BV at both V1 and V2.

Parameter (median)	Only BV-negative ^b	Unaltered BV status ^a
% virus entry, cervical CD4 T cells	2.31*	1.93*
Number of cervical CD4 T cells with detectable virus	1.03	1.02
% virus entry, blood CD4 T cells	1.36*	1.52*

*Significant change compared to baseline visit, at $p \leq 0.05$, as assessed by paired t-test.

^a n= 17 and 20, for cervical and blood comparisons, respectively.

^b n= 10 and 13, for cervical and blood comparisons, respectively.

Median fold changes were calculated first by calculation the corresponding ratio (e.g. V2/V1) for each participant, and then calculating a median of these ratios.

9. Supplemental Tables S6 and Figure S5. Why does one of these use the *S. mansoni* confirmed group as a comparator and the other use the Kato Katz confirmed group as a comparator?

Authors' response: Thank you for noticing this discrepancy. The *Sm* Serology/PCR group is our main comparator across the paper and Supplementary Fig 5 has now been amended to include this group.

10. Did (or could) the authors assess the frequency of a4b7 cells in the cervical mucosa? In concordance with their hypothesis that circulating systemic a4b7 cells would be elevated and would home to the genital mucosa, these would be expected to have been elevated as well.

Authors' response: While a4b7 can be quantified on CD4+ T cells derived from any site, its expression is downregulated after cell migration into mucosal sites (with upregulation of the molecule aEb7 that is important in tissue retention), and so it is of limited usefulness. Therefore, we opted not to assess this marker in the mucosal immunology panels.

11. Lines 176-8 – Why were only 3 women included in the RNA-Seq analysis when there were 12 who would have been eligible (if the criteria was CCA+KatoKatz+)? Was this subset pre-specified? Were these people who cleared their *S. mansoni* infection at V2 and V3? Did they have any co-infections? More detail is needed.

Authors' response: Our primary reason for the small subset included in the RNaseq subanalysis was cost (discussed now on Pg16, lines 5-9). This assay had not been included in our original study budget, but we felt that performance in a subset was important to confirm our unexpected results (i.e.: reduced mucosal HIV CD4+ susceptibility despite immune activation after *Sm* treatment). However, we believe that the paired before/after treatment approach that we took to this analysis means that this is a powerful tool despite the sample size (see Methods, Pg24), . These participants were randomly selected among the subset of participants who had *Sm* eggs demonstrated on stool microscopy. This is now clarified in the Results section (Pg11, lines 5-9) and more details about the participants age, schistosomiasis status etc are given in Suppl. Table 14.

12. Lines 350-1 – Who are the schistosomiasis-free volunteers used for the ex vivo CD4+ T cell phenotype and HIV entry study, and how were they chosen? Were they also the Canadian volunteers receiving praziquantel for prevention?

Authors' response: Thank you- we have now clarified this in the Methods section (Pg 22, lines 18-20). Specifically, these participants were Canadians without a recent history of travel to schistosomiasis-endemic regions. This group of individuals is different from the control group, which received praziquantel for prevention (described in point#3). This group consisted of four women and one man, who did not take praziquantel.

13. Please provide better justification for including all genes that have a $p < 0.10$ rather than $p < 0.05$ in your analysis. Is this something routinely done by other investigators?

Authors' response: We have now clarified several points pertaining to this analysis in the Results, Methods (Pg 24, lines 21-24) and Discussion (Pg16, lines 9-11), as follows. Since the goal of our analysis was to test a specific hypothesis about IFN pathway induction, rather than to characterize individual differentially-expressed genes, or DEGs (the usual goal of such studies), our criteria for candidate gene selection were less stringent. This strategy has been employed by other authors in similar contexts [e.g.: PMC5404932, <https://www.biorxiv.org/content/10.1101/426213v1>; in addition, $p < 0.1$ is the default threshold of the DESEQ2 package for identifying DEGs, PMID: 25516281].

Notably, recent work showed that RNA-seq meta-analysis is better at identifying true DEGs compared to any of the standard analysis pipelines [PMID: 24678608, REF#65 in the text]. In keeping with this, the threshold of $p = 0.1$ in our meta-analysis identifies genes that would have been considered differentially expressed at $p = 0.05$ by at least one of the five “gold standard” pipelines included in the meta-analysis (e.g. DESEQ2 or edgeR, see the Source Data file for raw meta-analysis output). Therefore, rather than restricting the analysis at the DEG selection step, we used two different approaches (Gene set enrichment and the IRG permutation test) to test the IFN pathway hypothesis on the meta-analysis-derived DEG set.

The results of both approaches were consistent and there was consistent overlap with our other experimental findings (i.e. cellular and cytokine data) suggesting that our RNA-seq analysis findings are biologically meaningful. However, given the hypothesis-testing (rather than exploratory) nature of the analysis, we agree with the reviewer that future studies will need to expand upon this work by including larger sample sizes and applying more stringent criteria for DEG selection to validate specific genes and gene networks associated with schistosomiasis and its treatment, as is now discussed in the Discussion section (pg15, lines 1-8) in the light of the recent work by Dupnik and colleagues (Refs# 44 & 45).

Minor comments:

Line 63 – the reference for the prevalence of *S. mansoni* infection is 15 years old – please use a more updated source.

Authors' response: The reference has been updated with the 2018 WHO citation.

Line 67 – I suggest being more precise with language. *S. haematobium* eggs are not “secreted” into the lumen of the gut; rather, they migrate there.

Authors' response: This has been corrected.

Line 87 – The fact that HIV positivity was an exclusion criterion is important,

since HIV infection could alter immune responses in women, so should also be mentioned early in the text (not just in the methods and Figure 1).

Authors' response: This has been done: we specified the HIV status of participants in the Abstract and Introduction as well as in the methods where appropriate.

Lines 245 – 247 – Please provide justification for including women in either the follicular or luteal phases of their menstrual cycles (rather than at a single time point in the menstrual cycle).

Authors' response: As described in point #2, we designed the study in order to ensure that all samples from each participant were collected at the same stage of the menstrual cycle, to control for well-described cyclical mucosal immune changes. However, since we had no *a priori* reason for thinking that Sm treatment would have more or less effect at different times of the cycle, we did not standardize menstrual phase at baseline (and permitted the enrollment of women who were not cycling).

Lines 268-269 – More details on the PCR used for *S. mansoni* is needed – such as primers, reaction, etc and/or “as previously described.”

Authors' response: Additional detail has been added (Pg 18, lines 2-5).

Lines 283-285 – More details on the cervical cell collection and preservation would be useful. Where was the processing performed? What percent viability was obtained? How many cells were obtained per cytobrush? Throughout the methods, please clarify where the testing was performed – at University of Toronto or UVRI or elsewhere.

Authors' response: This information has now been added to the Methods (Pg18, lines 19-24) and Supplementary Material (Suppl. Table 9- showing detailed statistics for cytobrush cell yields). The sites where different study procedures were performed are now clarified throughout the manuscript. All virus entry assays for trial participants were performed in real time using fresh (not frozen/thawed) samples on site at the UVRI-IAVI laboratory.

Figure 4 presents a large amount of data and the abbreviations for at least some things should be shown clearly – for example, at least the names of the top 10 signaling pathways and the genes pointed out on Fig 4a. Also, it would seem that any genes pointed out on Fig 4a should also be seen on Fig 4b.

Authors' response: We have now included the full names of the abbreviated genes and pathways appearing in Figure 4a/b in the respective Figure legend. Due to space limitations, only select genes from Fig4b could be shown in the volcano plot - this is also clarified in the Figure legend. More detailed data pertaining to this Figure (including interactive plots) can be found in the Source Data file.

Supplemental Table 2 – It would be useful to include the numbers of women that

were CCA 1+, 2+, and 3+ and the median number of eggs per gram detected with Kato Katz screening.

Authors' response: These data have been added, as requested, to Table 1 (former Suppl. Table 2).

Supplemental Figure 3 – Please show the numbers of patients included in each group and mention again here how you defined those with “high *S. mansoni* burden.”

Authors' response: These data have been added, as requested, in the figure (currently Supplemental Figure 2).

Supplemental Figure S8 – The figure is illegible.

Authors' response: We apologize for this oversight; the Figure has now been adjusted.

Reviewer #3 (Remarks to the Author)

This is a very interesting manuscript that provides convincing mechanistic evidence for how *Schistosoma mansoni* infections may increase a woman's susceptibility to HIV. This is a theory that has been around for over 20 years and there are studies that both support and contradict the idea. However, the majority of the work has been performed by schistosomiasis researchers with occasional support from HIV investigators. Therefore, addressing the question from an HIV perspective and using sophisticated techniques (e.g. the pseudovirus entry assay) not typically familiar to parasitologists is a welcome contribution. For the same reason, the limitations of the paper have to do with the authors' more limited understanding of schistosomiasis. However, these limitations are by no means deal breakers and can be easily addressed with some rewriting.

The primary example of this is the way the authors refer to the various infection groups. "High *S. mansoni* burden" has a very specific definition by the WHO: >400 eggs per gram of stool. While there may be 2-3 people in the study that meet this definition, using it to designate all those who were egg positive is inappropriate.

Rather than using "whole cohort", "confirmed *S. mansoni*", and "high *S. mansoni* burden", the authors should simply use CCA+, PCR/serology+, and Kato-Katz+. They have separately demonstrated that there are no *S. haematobium*-infected individuals in the cohort so the concept of "*S. mansoni* confirmed" is somewhat redundant. The change in group definitions should be changed throughout the manuscript and supplemental material.

Authors' response: We are grateful to the reviewer for these positive overall comments, and for their patience with our limited experience in the schistosomiasis field. We have now changed the group names as suggested throughout the main text, supplementary material and figures.

Were PCR and Kato-Katz tests performed at V2 and V3? It should be stated either way and the results should be included (how many previously positive became negative) if these tests were done.

Authors' response: The diagnostic testing algorithms are now clarified in the Results (pg5, lines 17-20)/ Methods (pg18, lines 1-11), and the more granular data describing the test results are provided in Supplementary material (Suppl. Tables 3, 4 & 6). Specifically, urine CCA and serology were performed at all visits, while PCR was performed at baseline and study closure and Kato-Katz was only performed at baseline.

There is a greater impact at V2 than V3 for several of the measures the authors used. Do the authors have any insight on whether the reduction in HIV infections susceptibility and increased IFN production is a transient or more lasting effect? Of course it has more public health impact if it is more permanent. Addressing

this question in the discussion (even if the answer is “more research”) would be helpful.

Authors’ response: We agree with the reviewer and are eager to address the important question of durability of the treatment effect as we move forward. The critical need for such additional work is now discussed in the Discussion section (pg13, lines 21-23).

Minor points/wording suggestions:

- 1) Line 51: use “Praziquantel treatment for schistosomiasis” rather than “Schistosomiasis treatment”
- 2) Line 60: add “for women with intestinal schistosomiasis” to the end of the sentences after “susceptibility”
- 3) Line 62: add “in Sm-endemic areas” to the end of the last sentence.
- 4) Line 66: change “egg-contaminated water” to “water containing infectious cercariae”
- 5) Line 70: add a sentence to the effect of “In addition, egg deposition induces a strong Th2 response that can cross regulate Th1 responses.”
- 6) Line 87: change “schistosomiasis” to “schistosome” The parasite causes the infection, schistosomiasis is the disease.

Authors’ response: We have made adjustments as suggested in points #1-6.

7) When describing the communities in Table S1, it would be informative to include a column denoting how far each village is from Lake Victoria. Also, what is the overall size of the study area? Alternatively, the authors could include a map of the study villages as a supplemental figure.

Authors’ response: That is an excellent suggestion, and we have now added a column to Supplementary Table 1 denoting the distance in km from the lake for each of the communities. The area of Entebbe is now given in the methods (pg16, line 3).

8) Line 97: replace “were deemed to have a high worm burden based on” with “had”

Authors’ response: This has been done.

9) Line 101: shouldn’t V1 and V2 be V2 and V3?

Authors’ response: Thank you for catching this typo- it has now been corrected.

10) Lines 104-106: include PCR and Kato Katz data for follow up time points if you have it.

Authors’ response: This has been done (Pg 6, lines 18-20)

11) Line 143: change “schistosomiasis treatment” to “praziquantel treatment”

Authors' response: This has been done.

12) Line 149: replace “a high Sm burden” with “being egg positive”

Authors' response: This has been done.

13) Figure 2: panels g and h are not labeled

Thank you for noting this- the figure has been amended.

14) Line 166: change infection designations.

Authors' response: This has been done.

15) Line 178: egg positive rather than high worm burden. Figure 4a is not mentioned in the text.

Authors' response: Thank you, the terminology has been changed, and Figure 4a is now cited in this section of the manuscript.

16) Line 222: Being egg positive rather than High Sm burden

Authors' response: This terminology has been corrected.

17) Line 651: change entire cohort to CCA+ and high worm burden to Kato-Katz+ (please check all text and figures—I have not necessarily noted every place it needs changing).

Authors' response: Thank you, we have gone through the manuscript and all accompanying related materials and changed the terminology as suggested.

REVIEWERS' COMMENTS:

Reviewer #1 (Remarks to the Author):

In this revised submission, the authors have been very responsive to reviewer comments and the manuscript has been significantly strengthened. This is an important and novel manuscript that will be of high interest to the readers of Nature Communications.

I do have a few remaining comments:

1. In the last sentence of the abstract, it makes more sense to refer to the infection as "S. mansoni infection" rather than calling it intestinal schistosomiasis, when much of your paper has focused on genital tract abnormalities in these women.
2. Figure 1 has an error between V1 and V2. It looks like a total of 7 women were excluded (36 enrolled - 29 seen 28 days later), yet 9 are shown and the chlamydia person seems to be repeated. Also in this figure, it looks like all people who were egg positive by Kato Katz were also either PCR or antibody positive (e.g., among the 24). Is this correct? It would be helpful to state this more clearly in the results section.
3. For Supplemental Table 11, please mention the cut-off OD value that is considered positive.
4. The authors' discussion of the FSH signaling pathway and its overlap with mTOR in the discussion is intriguing but weak. They have transcriptomic data before and after treatment; if they want to argue that the FSH signaling pathway appears to be upregulated after treatment not because of systemic changes in FSH but because of changes in the molecules that it shares with the mTOR pathway, it would be much more convincing if they could present data from their transcriptome analysis that showed that there were not changes in some of the gene expression that would be uniquely found in FSH signaling pathways but not in the mTOR pathway. If their preliminary data does not support this possibility, then this could be made clearer.

Reviewer #3 (Remarks to the Author):

The authors have done a nice job addressing the comments of the reviewers and improving the paper. My only additional comments are to make sure they notice a couple of typos:

--line 110, should be Sm rather than just m

--figure legend for supplementary figure 5, the asterisk in the figure is blue instead of red

Authors' response: We are again very thankful to the reviewers for their effort and valuable comments on our manuscript. Please kindly find our point-by-point responses to the reviewer's comments and suggestions below.

REVIEWERS' COMMENTS:

Reviewer #1 (Remarks to the Author):

In this revised submission, the authors have been very responsive to reviewer comments and the manuscript has been significantly strengthened. This is an important and novel manuscript that will be of high interest to the readers of Nature Communications.

I do have a few remaining comments:

1. In the last sentence of the abstract, it makes more sense to refer to the infection as "S. mansoni infection" rather than calling it intestinal schistosomiasis, when much of your paper has focused on genital tract abnormalities in these women. **Done**
2. Figure 1 has an error between V1 and V2. It looks like a total of 7 women were excluded (36 enrolled - 29 seen 28 days later), yet 9 are shown and the chlamydia person seems to be repeated.

The authors' response: This has been corrected.

Also in this figure, it looks like all people who were egg positive by Kato Katz were also either PCR or antibody positive (e.g., among the 24). Is this correct? It would be helpful to state this more clearly in the results section.

The authors' response: This has now been clarified in Results (Pg5, line 23).

3. For Supplemental Table 11, please mention the cut-off OD value that is considered positive.

The authors' response: Done.

4. The authors' discussion of the FSH signaling pathway and its overlap with mTOR in the discussion is intriguing but weak. They have transcriptomic data before and after treatment; if they want to argue that the FSH signaling pathway appears to be upregulated after treatment not because of systemic changes in FSH but because of changes in the molecules that it shares with the mTOR pathway, it would be much more convincing if they could present data from their transcriptome analysis that showed that there were not changes in some of the gene expression that would be uniquely found in FSH signaling pathways but not in the mTOR pathway. If their preliminary data does not support this possibility, then this could be made clearer.

The authors' response: Unfortunately, we are unable to find studies or databases containing detailed gene expression data specific to the effects of FSH and/or rapamycin on the transcriptome of blood-derived lymphocytes that can be applied to our study samples. Although there are studies on FSH effects in ovarian cells (PMID: 21996254) and studies on the effects of rapamycin treatment on cancer cell lines (PMCID: PMC5502825), it is difficult to derive clear FSH- versus mTOR pathway-associated signatures from these data that would apply to our PBMC RNA-seq analysis. We also do

not have samples available to directly test FSH levels as an alternative means to address the reviewer's concerns, but we are confident that our careful scheduling based on self-reported menstrual cycle stage should ensure that differences are not due to inter-visit differences in the menstrual cycle. We have now adjusted the discussion to this effect (pg14, lines 17-18).

Reviewer #3 (Remarks to the Author):

The authors have done a nice job addressing the comments of the reviewers and improving the paper. My only additional comments are to make sure they notice a couple of typos:

--line 110, should be Sm rather than just m **The authors' response: Done**

--figure legend for supplementary figure 5, the asterisk in the figure is blue instead of red **The authors' response: Done**